# One-Way Policy Optimization for Self-Evolving LLMs

Shuo Yang [* 1 2]   Jinda Lu [* 2]   Kexin Huang [2]   Chiyu Ma [2 3]   Shaohang Wei [1]   Yuyang Liu [1]   Guoyin Wang [2]
Jingren Zhou [4]   Li Yuan [1]

## Abstract

Reinforcement Learning with Verifiable Rewards (RLVR) has become a promising paradigm for scaling reasoning capabilities of Large Language Models (LLMs). However, the sparsity of binary verifier rewards often leads to low efficiency and optimization instability. To stabilize training, existing methods typically impose token-level constraints relative to a reference policy. We identify that such constraints penalize deviations indiscriminately; this can flip verifier-determined direction when the policy attempts to outperform the reference, thereby suppressing gains. To resolve this, we propose **One-Way Policy Optimization (OWPO)**, a method based on the principle of decoupling optimization direction from update magnitude. In OWPO, the verifier dictates the update direction, while the reference policy serves only to adjust the magnitude. Specifically, OWPO applies asymmetric reweighting: it performs **Accelerated Alignment** for inferior deviations (where the policy lags behind the reference) and **Gain Locking** for superior deviations (where the policy surpasses the reference). Furthermore, by incorporating iterative reference updates, OWPO creates a "Ratchet Effect" that continuously consolidates gains. Experimental results demonstrate that OWPO outperforms strong baselines, including DAPO, OPD, and MOPD, breaking the bottleneck of fixed priors to enable continuous self-evolution without reliance on external reference models.

## 1. Introduction

As large language models (LLMs) (Yang et al., 2025; Team et al., 2025; Comanici et al., 2025) are increasingly

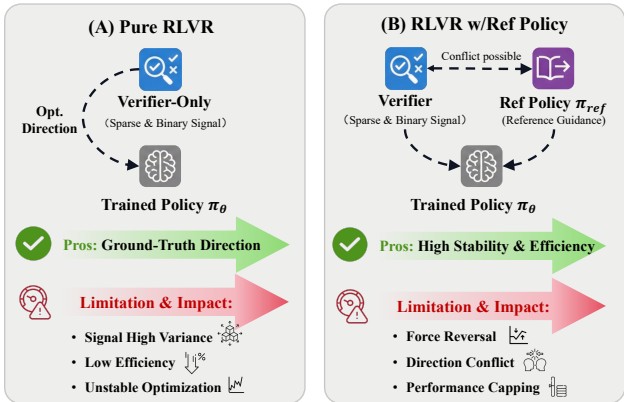

*Figure 1.* **Comparison of RLVR paradigms.** (A) **Pure RLVR** ensures the correct optimization direction via verifier signals but suffers from instability due to sparsity. (B) **RLVR with Reference Policy** (e.g., KL regularization) improves stability but introduces a **direction conflict**: the reference constraint can forcibly reverse reward-improving updates (Force Reversal) when the policy attempts to deviate from the prior, thereby capping performance.

trained for complex reasoning (Wei et al., 2022; Ma et al., 2026), Reinforcement Learning with Verifiable Rewards (RLVR) (Guo et al., 2025; Jaech et al., 2024; Meng et al., 2026) has become a promising post-training paradigm. RLVR leverages feedback from external verifiers to provide automatically checkable rewards, and has shown strong potential on verifiable reasoning tasks (Shao et al., 2024). However, verifier rewards are typically sparse and binary, and are often only available at the sequence level. This provides a weak training signal, reducing sample efficiency and frequently resulting in unstable optimization and slow convergence (Hochlehnert et al., 2025; Williams, 1992).

To address this, a common way to stabilize RLVR under sparse, binary verifier rewards is to introduce a reference policy and impose token-level guidance or constraints during optimization (Agarwal et al., 2024; Yang et al., 2025; Lightman et al., 2023; Huang et al., 2026). For example, on-policy distillation aligns the student distribution to a reference model on the student's own trajectories, while KL-regularized RL constrains policy updates by penalizing deviations from the reference distribution (Xu et al., 2025; Xiao et al., 2026). By providing dense token-level signals and keeping updates close to a known prior, these methods

*Equal contribution [1]Shenzhen Graduate School, Peking University [2]Qwen Pilot Team [3]Dartmouth College [4]Alibaba. Correspondence to: Guoyin Wang <guoyin.wang@alibaba-inc.com>, Li Yuan <yuanli-ece@pku.edu.cn>.

*Proceedings of the 43rd International Conference on Machine Learning*, Seoul, South Korea. PMLR 306, 2026. Copyright 2026 by the author(s).

*Table 1.* **Categorization of deviations.** We classify policy behaviors based on the verifier's signal (Advantage $A_t$) and the deviation from the reference ($\pi_\theta$ vs. $\pi_{\text{ref}}$). **Superior deviations** (Cases I & IV) indicate the policy outperforms the reference, whereas **Inferior deviations** (Cases II & III) indicate lag.

| Case | Adv. | Relation | Intuition | Deviation |
|:---:|:---:|:---:|:---|:---:|
| I | $A_t > 0$ | $\pi_\theta > \pi_{\text{ref}}$ | Correct & Confident | **Superior** |
| II | $A_t > 0$ | $\pi_\theta < \pi_{\text{ref}}$ | Correct but Hesitant | Inferior |
| III | $A_t < 0$ | $\pi_\theta > \pi_{\text{ref}}$ | Wrong & Over-confident | Inferior |
| IV | $A_t < 0$ | $\pi_\theta < \pi_{\text{ref}}$ | Wrong & Cautious | **Superior** |

substantially improve training stability and efficiency.

However, a key issue is **direction**: the token-level constraint pushes the policy toward the reference even when the reward-improving direction is away from it. In particular, KL regularization penalizes deviations from the reference regardless of whether they increase reward. For example, when a trajectory has positive advantage ($A > 0$) over the reference, the reward term would encourage moving further in that direction, but the KL term still exerts a strong pull-back toward the reference. If the KL pressure is large enough, it can **flip the effective update direction**, turning an improving step into a step back toward the reference and limiting gains beyond it (Figure 1(B)).

To resolve this conflict, we propose that the guidance from the reference policy should not interfere with the optimization direction determined by the verifier. Instead, we introduce a simple principle: **optimization direction and update magnitude should be decoupled**. Specifically, the verifier dictates the direction (the gradient sign), while the reference policy serves only to adjust the magnitude of the update. Building on this, we propose **One-Way Policy Optimization (OWPO)**. OWPO keeps the reward-improving direction unchanged, but uses the reference policy to scale the step size at the token level. This preserves stability while avoiding unnecessary suppression of superior deviations.

OWPO realizes the decoupling of optimization direction and update magnitude through token-level advantage reweighting. Specifically, based on the verifier's directional signal (the sign of the advantage) and the deviation from the reference model, we categorize the four interaction scenarios into two distinct regimes and apply specific weighting rules for each (as summarized in Table 1):

**(1) Accelerated Alignment** for Inferior Deviations (Cases II & III): We define an deviation as 'Inferior' when the current

policy $\pi_\theta$ lags behind the reference $\pi_{\text{ref}}$ along the verifier-determined direction—when $\pi_\theta$ is less confident than $\pi_{\text{ref}}$ on correct tokens or more confident on wrong ones. To counter this, OWPO increases the gradient weights ($w > 1$), leveraging the reference prior to accelerate correction.

**(2) Gain Locking** for Superior Deviations (Cases I & IV): We define an deviation as 'Superior' when $\pi_\theta$ surpasses $\pi_{\text{ref}}$ along the verifier-determined direction—exhibiting higher confidence on correct tokens or stronger suppression on wrong ones. To preserve these advanced gains, OWPO reduces the weights ($w < 1$), shielding them from high-variance interference while maintaining verifier's direction.

However, relying on a fixed $\pi_{\text{ref}}$ eventually creates a bottleneck. Since OWPO performs a one-way trust region update anchored to the reference, the optimization scope becomes constrained once $\pi_\theta$ significantly outperforms the initial prior. To transcend this limit, we extend OWPO into an iterative framework (Zelikman et al., 2022; Gulcehre et al., 2023). By periodically updating $\pi_{\text{ref}}$ to the optimized $\pi_\theta$, we establish a 'Ratchet Effect': each iteration consolidates recent gains into a new baseline, enabling sustained performance improvement across stages.

We propose **OWPO**, which decouples update magnitude from direction to prevent flipping the update direction. We show that its asymmetric reweighting preserves beneficial deviations, allowing OWPO to outperform strong baselines (e.g., MOPD) and break the "prior ceiling" for continuous self-evolution without reliance on external teacher models.

**Conflict of Interest Disclosure.** Several authors are employees of Alibaba Group, which developed the Qwen series of models used in this study. This work strictly utilizes the publicly accessible open-source weights of Qwen.

## 2. Preliminaries

In this section, we briefly review the frameworks that form the basis of our study, progressing from RLVR methods to reference-guided strategies.

**Group Relative Policy Optimization (GRPO)** (Shao et al., 2024). GRPO eliminates value networks by estimating advantages via group sampling, typically incorporating a KL penalty towards $\pi_{\theta_{\text{old}}}$. Given a group of outputs $\{y_i\}_{i=1}^G$ sampled from $\pi_{\theta_{\text{old}}}$, the advantage is normalized against group statistics:

$$\hat{A}_i = \frac{R_i - \text{mean}(\{R_k\}_{k=1}^G)}{\text{std}(\{R_k\}_{k=1}^G)}, \text{ where } R_i = \mathbb{I}(\text{Verify}(x, y_i)). \tag{1}$$

**Decoupled Clip and Dynamic sAmpling Policy Optimization (DAPO)** (Yu et al., 2025) refines GRPO by removing the standard KL penalty to prevent over-regularization and introducing asymmetric clipping ($1 - \epsilon_{\text{low}}, 1 + \epsilon_{\text{high}}$). It

stabilizes gradients by enforcing a *dynamic sampling* condition (ensuring mixed success/failure within a batch) and maximizes the following token-normalized objective:

$$\mathcal{J}_{\text{DAPO}}(\theta) = \mathbb{E}_{\substack{x\sim\mathcal{D} \\ \{y_i\}_{i=1}^{G}\sim\pi_{\theta_{\text{old}}}}} \left[ \frac{1}{Z} \sum_{i=1}^{G} \sum_{t=1}^{|y_i|} \min\Big( r_{i,t}(\theta)\hat{A}_{i,t}, \right.$$
$$\left. \text{clip}(r_{i,t}(\theta), 1-\epsilon_{\text{low}}, 1+\epsilon_{\text{high}})\hat{A}_{i,t} \Big) \right]. \tag{2}$$

To address the variance of sparse rewards, reference-guided methods provide dense supervision. **On-Policy Distillation (OPD)** (Agarwal et al., 2024) minimizes the reverse KL divergence $\mathbb{D}_{\text{KL}}(\pi_\theta || \pi_{\text{ref}})$ on student-generated trajectories. Extending this, **Multi-teacher On-Policy Distillation (MOPD)** (Xiao et al., 2026) formulates distillation as an RL problem, optimizing a weighted objective:

$$\mathcal{J}_{\text{MOPD}}(\theta) = \mathbb{E}_{\substack{x\sim\mathcal{D} \\ y\sim\mu_\theta(\cdot|x)}} \left[ \frac{1}{|y|} \sum_{t=1}^{|y|} r_t(\theta) \cdot \hat{A}_{\text{MOPD},t} \cdot \log \pi_\theta(y_t|x, y_{<t}) \right] \tag{3}$$

MOPD constructs a hybrid advantage that combines the dense teacher log-ratios with sparse outcome rewards:

$$\hat{A}_{\text{MOPD},t} = \underbrace{\text{sg}\left[ \log \frac{\pi_{\text{ref}}(y_t|\cdot)}{\pi_\theta(y_t|\cdot)} \right]}_{\text{Distillation Advantage}} + \alpha \hat{A}_{\text{ORM}}, \tag{4}$$

where $\text{sg}[\cdot]$ is the stop-gradient operator. While effective, this hybrid formulation can introduce conflicting gradients when the verification signal opposes the reference prior.

## 3. Methodology

To address the sparsity of rewards in RLVR, we propose **One-Way Policy Optimization (OWPO)** , which integrates two complementary signals: the **verifier's feedback** (acting as a proxy for ground truth) and the **reference policy** (providing a prior for token-level confidence). OWPO is built on a core principle: the **optimization direction** should be dictated by the verifier, while the **update magnitude** should be modulated by the reference policy.

### 3.1. One-Way Policy Optimization

**Directional Deviation.** Formally, let $\pi_\theta$ denote the current policy. Given an input $x$, $\pi_\theta$ generates a trajectory $y = (y_1, \ldots, y_T)$. A verifier evaluates this trajectory to compute the advantage $A_t$. Concurrently, the reference policy $\pi_{\text{ref}}$ computes the token-level log-probability $\log \pi_{\text{ref}}(y_t \mid s_t)$ on the data generated by $\pi_\theta$ (i.e., on-policy data), a baseline confidence for each token. To quantify how the policy performs relative to the reference along the verifier's direction, we define the **Directional Deviation** as:

$$\delta_t(\theta) \triangleq \text{sgn}(A_t) \cdot \log \frac{\pi_\theta(y_t \mid s_t)}{\pi_{\text{ref}}(y_t \mid s_t)}. \tag{5}$$

Here, $\delta_t$ combines the correctness (sign of $A_t$) and the relative confidence (log-ratio) into a single metric. Crucially, a positive $\delta_t$ corresponds to a **Superior deviation** (where the policy is "ahead" of the reference in the correct direction), while a negative $\delta_t$ indicates an **Inferior deviation**.

**The One-Way Objective.** To implement the decoupling of optimization direction and update magnitude, we apply a dynamic scalar weight $w_t$ to the standard PPO objective. OWPO maximizes the following reweighted objective:

$$\mathcal{J}_{\text{OWPO}}(\theta) = \mathbb{E}\left[ \sum_t w_t \cdot \min\Big( r_t(\theta) A_t, \right.$$
$$\left. \text{clip}\big( r_t(\theta), 1-\epsilon, 1+\epsilon \big) A_t \Big) \right] \tag{6}$$

where $r_t(\theta)$ denotes the standard importance sampling ratio, and the one-way dynamic weight is defined as:

$$w_t \triangleq \text{sg}[\text{clip}(\exp(-\delta_t), \epsilon_{\text{low}}, \epsilon_{\text{high}})]. \tag{7}$$

where $\text{sg}[\cdot]$ denotes the stop-gradient operator.

**Mechanism Analysis.** The sign of $\delta_t$ determines the weight $w_t$, implementing the asymmetric one-way strategy:

- **Accelerated Alignment** ($\delta_t < 0 \Rightarrow w_t > 1$): This occurs during an **Inferior deviation**, where the policy lags behind the reference (e.g., lower confidence on correct tokens or higher confidence on wrong ones). Here, $w_t > 1$ amplifies the gradient, leveraging the reference prior to accelerate the alignment process.
- **Gain Locking** ($\delta_t > 0 \Rightarrow w_t < 1$): This occurs during a **Superior deviation**, where the policy outperforms the reference (e.g., higher confidence on correct tokens). Here, $w_t < 1$ reduces the gradient weight. This shrinks the update step size to lower variance, effectively "locking" these sparse but valuable gains into the policy and protecting them from instability.

### 3.2. Iterative Bootstrapping and Algorithm

While OWPO effectively optimizes within a one-way trust region, relying on a fixed $\pi_{\text{ref}}$ eventually creates a bottleneck. As the policy $\pi_\theta$ significantly improves, the initial reference becomes an outdated lower bound.

**Self-Evolution via Ratchet Effect.** To break this ceiling and achieve continuous self-evolution, we extend OWPO into an iterative bootstrapping process (Figure 2). The core mechanism involves a "Hard Swap" of the reference policy at the end of each iteration: $\pi_{\text{ref}}^{(k+1)} \leftarrow \pi_\theta^{(k)}$. This updates the baseline and re-calibrates $\delta_t$, creating a "Ratchet Effect" that supports a unidirectional climb toward higher rewards.

**Implementation Pipeline.** In addition to the iterative update, we employ a dynamic curriculum for sample selection

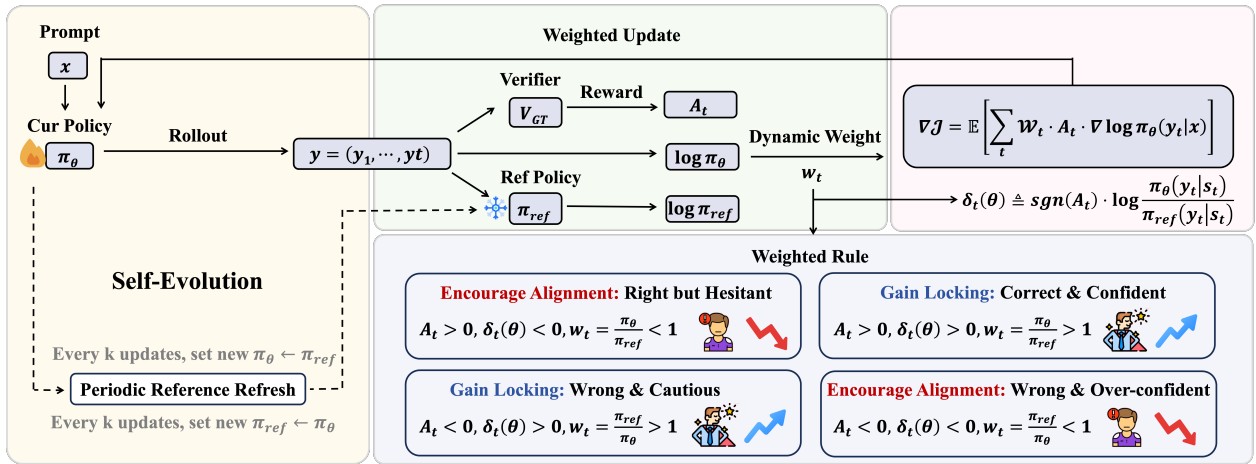

*Figure 2.* Overview of OWPO. The pipeline decouples the optimization direction (determined by the Verifier $V_{GT}$) from the update magnitude (modulated by the Ref Policy $\pi_{\text{ref}}$). Based on the Directional Deviation $\delta_t$, OWPO dynamically applies asymmetric weights $w_t$: executing Accelerated Alignment for inferior deviations to correct lag, and Gain Locking for superior deviations to protect exploration gains. Furthermore, the Periodic Reference Refresh mechanism (bottom left) iteratively updates $\pi_{\text{ref}}$ to sustain continuous self-evolution.

---

**Algorithm 1** Self-Evolving RLVR with OWPO

1: **Input:** Initial policy $\pi_\theta$, dataset $\mathcal{D}$, verifier $\mathcal{V}$, group size $G$, update interval $K$.
2: Initialize reference policy $\pi_{\text{ref}} \leftarrow \pi_\theta$.
3: **repeat**
4:   Sample prompts $x \sim \mathcal{D}$ and generate rollouts $\mathbf{y} = \{y_1, \ldots, y_G\} \sim \pi_\theta(\cdot|x)$.
5:   Compute rewards via $\mathcal{V}$ and estimate advantages $\hat{A}$.
6:   Calculate active count $N_{\text{act}}$ linearly decaying from $G \rightarrow 1$ within current stage.
7:   **for** each sample $y_i$ in $\mathbf{y}$ **do**
8:     **if** $y_i$ is in top-$N_{\text{act}}$ samples based on $|\hat{A}|$ **then**
9:       Compute token-level weights $w_{i,t}$.
10:     **else**
11:       Set standard weights $w_t \leftarrow 1$.
12:     **end if**
13:   **end for**
14:   Update $\pi_\theta$.
15:   **if** current step mod $K$ is 0 **then**
16:     Update reference policy: $\pi_{\text{ref}} \leftarrow \pi_\theta$.
17:   **end if**
18: **until** convergence

within each training stage. Specifically, the number of active samples within a group that participate in the reweighting calculation (Active Sample Count) linearly decays from $G$ to 1 as training progresses. This strategy gradually shifts focus from broad exploration to exploitation. The complete training procedure is detailed in Algorithm 1.

## 4. Theoretical Analysis

In this section, we analyze the theoretical properties of OWPO by connecting it to asymmetric directional regularization and examining its effective progress dynamics. OWPO is implemented with a PPO/DAPO-style clipped surrogate, while our analysis focuses on the **local first-order regime** at the start of an update, where $\pi_\theta \approx \pi_{\text{old}}$ and thus $r_t(\theta) \approx 1$. In this regime, the clipped surrogate reduces to a weighted policy-gradient direction, allowing us to isolate how OWPO modulates the verifier-induced update. Thus, the regularization view below is local rather than a global identity for the full clipped objective.

Recalling the directional deviation $\delta_t(\theta)$ and the reweighting coefficient $w_t$ in Eqs. 5 and 7, we first characterize the optimization landscape of OWPO through this local equivalence.

**Lemma 4.1** (Local Directional Regularization)**.** *In the local first-order regime described above, the OWPO update direction $\nabla J_{\text{OWPO}}(\theta) = \mathbb{E}_{\tau \sim \pi_{\theta_{\text{old}}}}[\sum_t w_t A_t \nabla_\theta \log \pi_\theta(y_t \mid s_t)]$ is equivalent to the gradient of a directionally regularized policy-gradient objective $\nabla_\theta(J_{\text{PG}}(\theta) + \Omega(\theta))$, where the regularization term is given by:*

$$\Omega(\theta) = \mathbb{E}_{\tau \sim \pi_{\theta_{\text{old}}}}\left[\sum_t |A_t| \cdot \psi(\delta_t(\theta))\right], \qquad (8)$$

*with the potential function $\psi(\delta) = \int_0^\delta (w(u) - 1)\, du$.*

Unlike standard KL divergence, the regularization $\Omega(\theta)$ incorporates $\text{sgn}(A_t)$ via $\delta_t$, rendering it **direction-aware**. Based on Lemma 4.1, the total gradient can be expressed as

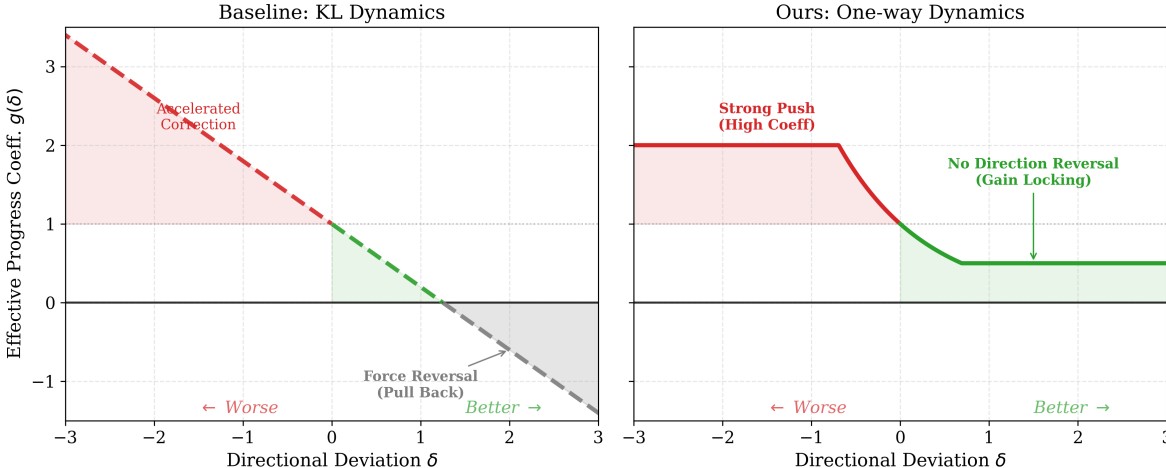

*Figure 3.* **Linearized effective progress dynamics. Left:** Standard KL exhibits **Force Reversal** ($g < 0$) when deviations are large, effectively negating the reward signal. **Right:** OWPO maintains Unidirectional Dynamics ($g \geq \epsilon_{\text{low}}$). The asymmetric profile enables **Accelerated Alignment** for lags ($\delta < 0$) and **Variance Reduction** for gains ($\delta > 0$), preventing any direction flip.

a scaled version of the standard policy-gradient estimator:

$$\nabla_\theta\big(J_{\text{PG}} + \Omega\big) = \mathbb{E}_{\tau \sim \pi_{\theta_{\text{old}}}}[\textstyle\sum_t g_{\text{OWPO}}(\delta_t)\, A_t \nabla_\theta \log \pi_\theta(y_t \mid s_t)] \tag{9}$$

We define the *Effective Progress Coefficient* as

$$g_{\text{OWPO}}(\delta) \triangleq 1 + \psi'(\delta) \equiv w(\delta).$$

This local view also clarifies the contrast with symmetric KL regularization. Let

$$z_t(\theta) = \log \frac{\pi_\theta(y_t \mid s_t)}{\pi_{\text{ref}}(y_t \mid s_t)}, \qquad \delta_t = \text{sgn}(A_t)z_t.$$

A local quadratic approximation of a symmetric KL-style penalty gives the token-level surrogate

$$\ell_{\text{KL}}(z_t) = A_t z_t - \frac{\beta}{2}z_t^2.$$

For $A_t \neq 0$, its gradient can be written as

$$\nabla_\theta \ell_{\text{KL}} = \left(1 - \frac{\beta}{|A_t|}\delta_t\right) A_t \nabla_\theta \log \pi_\theta(y_t \mid s_t).$$

Thus, when $\delta_t > |A_t|/\beta$, the effective coefficient becomes negative, causing **Force Reversal**: the KL term locally flips the verifier-induced update back toward the reference (Figure 3, Left). In contrast, OWPO uses $g_{\text{OWPO}}(\delta) = w(\delta) > 0$, so the reference policy only modulates update magnitude and the verifier-determined direction is preserved.

**Proposition 4.2** (Properties of One-way Dynamics)**.** *Assuming $0 < \epsilon_{\text{low}} < 1 < \epsilon_{\text{high}}$, the coefficient $g_{\text{OWPO}}(\delta)$ satisfies:*

*1. **Non-reversal:** $\forall \delta \in \mathbb{R},\ g_{\text{OWPO}}(\delta) \geq \epsilon_{\text{low}} > 0.$*

*2. **Asymmetry:** $g_{\text{OWPO}}(\delta) > 1$ for $\delta < 0$ (Inferior), and $g_{\text{OWPO}}(\delta) \in [\epsilon_{\text{low}}, 1)$ for $\delta > 0$ (Superior).*

Proposition 4.2 highlights two distinct regimes aligned with our design: (1) For **Inferior deviations** ($\delta < 0$), $g > 1$ creates a steep gradient barrier to enforce **Accelerated Alignment**; (2) For **Superior deviations** ($\delta > 0$), the decay of $g$ towards $\epsilon_{\text{low}}$ prevents direction reversal while strictly reducing the step size (Figure 3, Right). This naturally leads to variance reduction during the exploration phase.

**Local forward/reverse-KL view.** OWPO is not an exact optimizer of either a forward-KL or reverse-KL objective. Nevertheless, its unclipped weights provide a useful local KL-style interpretation. When $A_t > 0$, $w_t = \pi_{\text{ref}}(y_t \mid s_t)/\pi_\theta(y_t \mid s_t)$, which amplifies positive-advantage tokens where the current policy lags behind the reference and down-weights tokens where it already surpasses the reference. This resembles a conservative reverse-KL-like correction. When $A_t < 0$, $w_t = \pi_\theta(y_t \mid s_t)/\pi_{\text{ref}}(y_t \mid s_t)$, which amplifies the suppression of over-confident wrong tokens and reduces updates on already cautious tokens, giving a forward-KL-like correction. This view is complementary to, rather than a replacement for, the directional regularization analysis above.

**Corollary 4.3** (Variance Bounding)**.** *Let $G_t = A_t \nabla_\theta \log \pi_\theta(y_t \mid s_t)$ be the raw gradient estimator. The second moment of the weighted gradient satisfies:*

$$\epsilon_{\text{low}}^2\, \mathbb{E}[\|G_t\|^2] \leq \mathbb{E}[\|w_t G_t\|^2] \leq \epsilon_{\text{high}}^2\, \mathbb{E}[\|G_t\|^2]. \tag{10}$$

*Specifically, conditioned on **Superior deviations** ($\delta > 0$), we have $\mathbb{E}[\|w_t G_t\|^2 \mid \delta > 0] \leq \mathbb{E}[\|G_t\|^2 \mid \delta > 0].$*

Corollary 4.3 formalizes the **Gain Locking** mechanism: by contracting the gradient's second moment in high-reward

regions (Superior deviations), OWPO stabilizes the retention of sparse, high-value signals against stochastic noise.

# 5. Experiments

## 5.1. Experimental setup

**Implementation Details.** We conduct experiments on two base models: Qwen-2.5-Math-7B-Base (Hui et al., 2024; Yang et al., 2024) (8k context) and Qwen-3-8B-Base (Yang et al., 2025) (20k context). All models are optimized in the RLVR post-training setting using the DAPO-Math-17k dataset (Yu et al., 2025), with sparse, verifiable, sequence-level feedback. We use the AdamW optimizer with a learning rate of $1e-6$ and a global batch size of 512. Rollout generation is accelerated via vLLM (Kwon et al., 2023), and outcomes are verified using a mathematical verifier.

**Reference Policy.** To ensure a rigorous comparison, we adopt a two-stage protocol for all reference-guided baselines (i.e., OPD, MOPD, and OWPO). We first train a standard DAPO baseline and evaluate it periodically. The checkpoint achieving the highest Pass@1 score on the AIME24 validation set is selected as the fixed reference policy $\pi_{\text{ref}}$. This ensures $\pi_{\text{ref}}$ represents a strong, converged policy rather than a weak initialization, posing a substantial challenge for further improvement.

**Baseline Descriptions.** We compare OWPO against representative RLVR and distillation methods: **GRPO** (Shao et al., 2024) and **DAPO** (Yu et al., 2025) (pure RLVR); **Sym-DAPO**[1], which adds a symmetric KL penalty to DAPO to serve as a control for direction-agnostic constraints; **OPD** (Agarwal et al., 2024) (on-policy reverse KL minimization); and **MOPD** (Xiao et al., 2026) (hybrid distillation with sparse rewards). Extensive hyperparameter tuning was applied to all baselines (see Appendix B).

**Evaluation Protocol.** We employ AIME24 as the validation set for monitoring training dynamics and checkpoint selection. For each method, we report results at two distinct stages: (i) the **best** checkpoint based on AIME24 performance, and (ii) the **converged** checkpoint after training stabilizes. Crucially, for any given method, the benchmark scores reported in Table 2 (including AIME25 and AMC) are evaluated using the **same checkpoint determined by AIME24**, ensuring a consistent cross-benchmark comparison. We report Pass@1 and Pass@16 accuracy and additionally evaluate general-domain transfer on GPQA and MMLU-Pro (Wang et al., 2024).

---

[1]Sym-DAPO adapts (Xu et al., 2025) by replacing GRPO with the stronger DAPO while maintaining symmetric constraints.

## 5.2. Main results

**Performance on Math Benchmarks.** As shown in Table 2, OWPO consistently outperforms all baselines across both base models. While reference-guided baselines (OPD, MOPD) exhibit high stability with minimal gaps between **best** and **converged** scores, they struggle to significantly surpass the reference. MOPD achieves only marginal average gains ($+0.93\%$ and $+1.13\%$), suggesting that symmetric KL penalties induce **Force Reversal**, constraining the model to the reference distribution even when it attempts to improve. In contrast, OWPO effectively breaks this performance ceiling. By leveraging the one-way mechanism to apply **Accelerated Alignment** for Inferior deviations while performing **Gain Locking** for Superior deviations, OWPO achieves substantial average improvements ($+3.88\%$ and $+7.71\%$). Tables 5 and 6 show OWPO maintains competitive Pass@16 results, validating reliability. Notably, OWPO maintains superior performance at convergence, avoiding the regression typical of pure RL (e.g., GRPO, DAPO).

**Training Dynamics.** The learning curves in Figure 4 further highlight the optimization dynamics. Reference-guided methods (OPD, MOPD, OWPO) demonstrate a rapid initial performance surge compared to the pure exploration method (DAPO), validating the benefit of dense supervision. However, a clear divergence emerges in the later stages: Pure distillation (OPD) fluctuates around the reference performance; MOPD exceeds the reference but plateaus due to the regularization conflict. OWPO, conversely, maintains a continuous upward trajectory, effectively decoupling the optimization direction from the reference constraints and achieving a significantly higher convergence point.

**Generalization Capabilities.** We evaluate generalization on general science benchmarks (GPQA and MMLU-Pro) in Table 3. Despite training strictly on mathematical data, OWPO demonstrates superior generalization, significantly outperforming MOPD and DAPO. This indicates that the reasoning patterns reinforced by OWPO are robust and transferable, rather than overfitted to mathematical templates.

## 5.3. Transcending Suboptimal Priors

To enable *safe self-evolution* despite imperfect priors, we initialize $\pi_{\text{ref}}$ with a suboptimal DAPO checkpoint (30.0% on AIME24) rather than the best one (37.19%). This rigorously tests if the algorithm can *transcend* a weak prior without being constrained by the "prior ceiling."

**Iterative Self-Evolving Protocol.** We compare OWPO against MOPD (the strongest baseline) using an identical iterative protocol. At the end of each stage, $\pi_{\text{ref}}$ is updated via a "hard swap" to the best-performing checkpoint within that stage ($\pi_{\text{ref}} \leftarrow \pi_{\theta}^{\text{best}}$). This ensures both methods benefit from the same "ratchet" schedule, isolating performance

*Table 2.* **Performance on math benchmarks.** We compare OWPO against baselines on Qwen-2.5 and Qwen-3 base models. Results report Pass@1 accuracy evaluated using $k = 32$ **samples** per problem (Average@32) at both best and converged cpts. The table contrasts **Pure RLVR** methods (top two rows) with those incorporating a **Reference Policy** (bottom four rows).

| Model | Method | AIME24 | | AIME25 | | AMC | | Average | |
|---|---|---|---|---|---|---|---|---|---|
| | | best | converge | best | converge | best | converge | best | converge |
| Qwen-2.5-Math-7B-Base | GRPO | 32.08 | 29.79 | 11.77 | 12.60 | 67.39 | 64.87 | 37.08 | 35.75 |
| | DAPO | 37.19 | 31.88 | 15.62 | **18.75** | 68.86 | **77.86** | 40.56 | 42.83 |
| | Sym-DAPO | 35.00 | 32.81 | 16.04 | 15.63 | 67.58 | 71.58 | 39.54 | 40.01 |
| | MOPD | 38.85 | 37.19 | 15.93 | 16.98 | 68.56 | 62.91 | 41.11 | 39.03 |
| | OPD | 38.12 | 35.52 | 15.21 | 16.56 | 68.59 | 69.05 | 40.64 | 40.38 |
| | OWPO | **41.67** | **39.90** | **17.92** | 18.44 | **72.14** | 74.02 | **43.91** | **44.12** |
| Qwen-3-8B-Base | GRPO | 31.67 | 29.27 | 23.54 | 20.42 | 67.73 | 66.98 | 40.98 | 38.89 |
| | DAPO | 36.67 | 34.58 | 27.39 | 30.63 | 71.88 | 73.91 | 45.31 | 46.37 |
| | Sym-DAPO | 36.04 | 33.23 | 26.88 | 24.79 | 70.74 | 69.95 | 44.55 | 42.66 |
| | MOPD | 37.80 | 36.04 | 29.38 | 27.19 | 72.03 | 72.82 | 46.40 | 45.35 |
| | OPD | 38.23 | 35.42 | 27.81 | 26.56 | 71.16 | 70.22 | 45.73 | 44.07 |
| | OWPO | **44.38** | **42.19** | **33.54** | **31.67** | **77.79** | **78.36** | **51.90** | **50.74** |

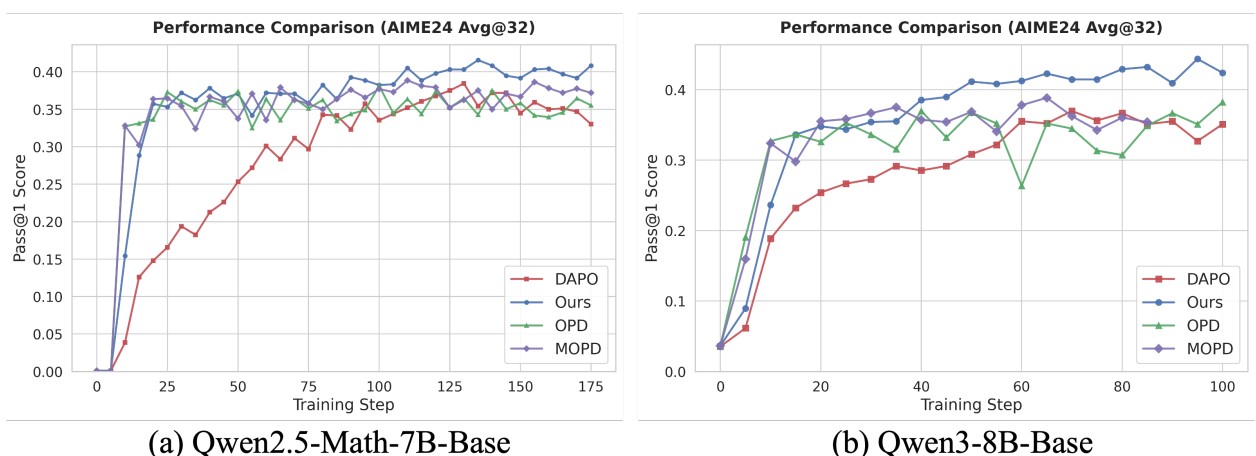

(a) Qwen2.5-Math-7B-Base  (b) Qwen3-8B-Base

*Figure 4.* **Training Dynamics Comparison.** The Pass@1 score curves on AIME24 during training. While distillation-based methods (OPD, MOPD) converge quickly, they tend to plateau near the reference performance. OWPO exhibits continuous improvement, significantly outperforming the baselines at convergence.

differences to the underlying optimization dynamics.

**Results.** As shown in Figure 5, both methods successfully break the performance ceiling of the initial weak reference via iterative bootstrapping. However, a significant divergence in the efficiency of self-evolution is observed. MOPD exhibits a slower rate of improvement, requiring **6 iterations** to reach a saturation point of approximately **37%**. In contrast, OWPO maintains a steeper upward trajectory and converges to a superior Pass@1 score of $\approx$ **40%** within only **4 iterations**. This empirical evidence confirms that while symmetric distillation (MOPD) provides stability, it inherently limits the magnitude of **Superior deviations**. OWPO's one-way mechanism effectively "locks in" these

deviations, enabling the model to transcend the suboptimal prior more rapidly and thoroughly.

### 5.4. Ablation Studies

To rigorously verify the necessity of the proposed asymmetric trust region, we adhere to the same *suboptimal reference* setting as in Sec. 5.3 but perform single-stage training to isolate the mechanical contributions. We compare the full OWPO against three variants:

- **w/o Asym (Symmetric Reweighting):** Removes the dependency on the advantage sign. Weights are determined solely by the magnitude of deviation $|\delta|$, applying sym-

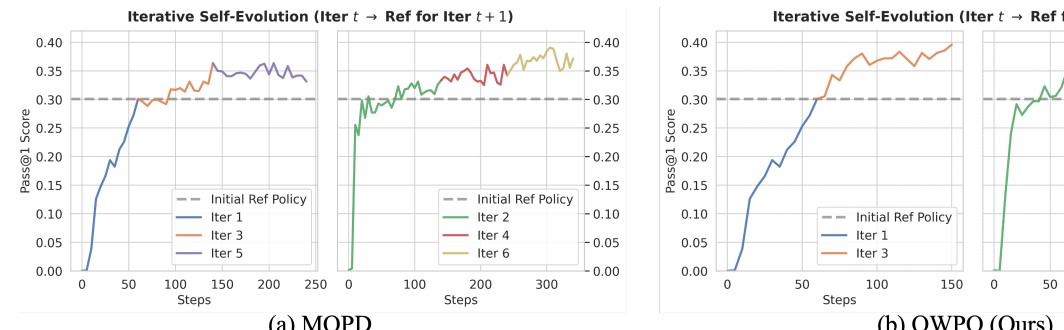

*Figure 5.* **Iterative Self-Evolution.** Comparison of **MOPD (left)** and **OWPO (right)** starting from a suboptimal prior ($\approx 30\%$, grey dashed line). We employ a stage-wise bootstrapping protocol: at the end of each iteration, the best checkpoint is frozen and serves as the $\pi_{\mathrm{ref}}$ for the subsequent iteration (e.g., the final model of Iter 1 becomes $\pi_{\mathrm{ref}}$ for Iter 2). OWPO demonstrates superior efficiency, reaching $\approx 40\%$ accuracy in just 4 iterations, whereas MOPD saturates at a lower $\approx 37\%$ accuracy despite requiring 6 iterations. This means that OWPO not only transcends the prior more effectively but also achieves better performance with reduced computational cost and time.

*Table 3.* **Results on general science benchmarks.** Comparison of different baselines on GPQA and MMLU-Pro.

| Model | Method | GPQA | MMLU-Pro |
|---|---|---|---|
| Qwen-2.5-Math-7B-Base | GRPO | 34.34 | 36.94 |
| | DAPO | 39.27 | 44.09 |
| | Sym-DAPO | 37.25 | 42.93 |
| | MOPD | 38.76 | 43.81 |
| | OPD | 39.39 | 43.34 |
| | OWPO | **41.03** | **46.85** |
| Qwen-3-8B-Base | GRPO | 50.44 | 64.71 |
| | DAPO | 52.27 | 65.29 |
| | Sym-DAPO | 50.12 | 65.58 |
| | MOPD | 50.51 | 65.99 |
| | OPD | 52.27 | 65.91 |
| | OWPO | **54.55** | **67.41** |

metric penalties regardless of whether the deviation is **Superior** or **Inferior**.

- **w/o Locking (Only Alignment):** Retains Accelerated Alignment for **Inferior deviations** ($w > 1$) but disables Gain Locking for Superior ones ($w = 1$ when $\delta > 0$).
- **w/o Accel (Only Gain Locking):** Retains Gain Locking for **Superior deviations** ($w < 1$) but removes Accelerated Alignment (setting $w = 1$ when $\delta < 0$).

**Analysis of Asymmetry.** Figure 6 reveals that enforcing symmetric constraints causes severe performance regression. The **w/o Asym** variant suffers significant drops on both AIME24 and AIME25. Notably on AIME25, symmetric reweighting yields a marginal gain ($+0.9\%$) compared to OWPO's fourfold improvement ($+3.8\%$). This confirms that symmetric regularization induces a "force-reversal" effect, erroneously penalizing the beneficial explorations necessary to outperform an imperfect reference.

**Analysis of Components.** Figure 6 shows both mechanisms

prove indispensable. **w/o Locking** underperforms OWPO significantly, indicating that **Gain Locking** is crucial for retaining sparse high-reward signals (i.e., preserving Superior deviations). Similarly, **w/o Accel** fails to match the full method, suggesting that **Accelerated Alignment** is vital for efficiently pruning incorrect paths (i.e., correcting Inferior deviations). In summary, OWPO achieves optimal gains only by synergizing these two asymmetric mechanisms.

## 6. Related Work

**RLVR for Reasoning Scaling.** Recent reasoning-oriented post-training has increasingly adopted reinforcement learning with verifiable rewards (RLVR) (Guo et al., 2025; Jaech et al., 2024; Yang et al., 2026), where external verifiers (e.g., code executors or math verifier) provide outcome-level supervision. The OpenAI o1 (Jaech et al., 2024) series demonstrated a viable pathway for reinforcing long-chain reasoning behaviors through large-scale RL. Subsequently, DeepSeek-R1 (Guo et al., 2025) employed GRPO (Shao et al., 2024) to scale training using solely verifiable signals, maintaining stability and continuous improvement through mechanisms. To address the stability challenges (Schulman et al., 2015) inherent in PPO (Schulman et al., 2017) and GRPO, subsequent research (Chen et al., 2025) has largely evolved around modifications to "trust regions and clipping forms." For instance, DAPO (Yu et al., 2025) mitigates training instability via decoupled clipping and dynamic sampling, while GSPO (Zheng et al., 2025) elevates importance ratios and clipping from the token level to the sequence level.

**Distillation Meets RL.** Another coherent line of research leverages distillation to provide dense token-level signals, thereby enhancing sample efficiency and stability. On-Policy Distillation (Lu & Thinking Machines Lab, 2025) (also known as GKD (Agarwal et al., 2024)) introduces teacher feedback on sequences generated by the student policy to mitigate distribution mismatch, validated by

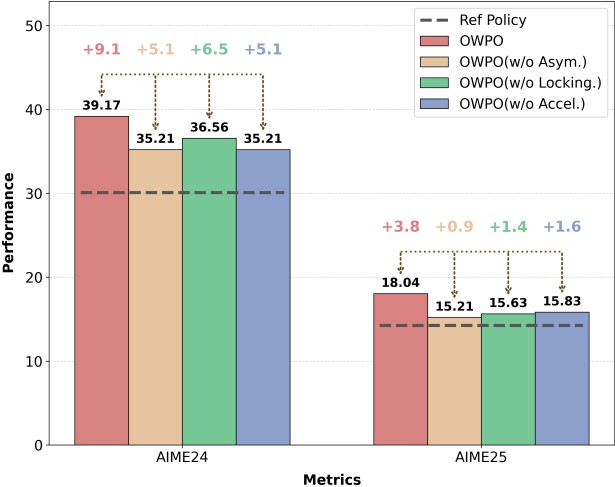

*Figure 6.* **Ablation study under the suboptimal reference setting.** We report the Pass@1 accuracy on AIME24 and AIME25 benchmarks. The grey dashed line indicates the baseline performance of the reference policy. We compare the full OWPO against variants removing the asymmetric design (`w/o Asym`), Gain Locking (`w/o Locking`), or Accelerated Alignment (`w/o Accel`).

Qwen3 (Yang et al., 2025) as a standard recipe. Moving toward tighter integration, KDRL (Xu et al., 2025) unifies KD and RL into a single objective; MiMo-V2-Flash introduces MOPD (Xiao et al., 2026), a multi-teacher framework utilizing RL expert teachers for dense guidance; and SCoRe (Lyu et al., 2025) combines correction distillation with short-term RL. Unlike regression or symmetric KL approaches, OWPO uses the reference strictly to modulate update magnitude, establishing a direction-aware one-way trust region that prioritizes the verifier's guidance for sustainable self-evolution.

## 7. Conclusion

In this paper, we introduced **One-Way Policy Optimization (OWPO)**, a framework that resolves the stability-exploration conflict in RLVR by decoupling optimization direction from update magnitude. By constructing a direction-aware trust region, OWPO applies **Accelerated Alignment** to correct inferior deviations while executing **Gain Locking** to preserve superior deviations, effectively preventing the **Force Reversal** effect inherent in symmetric KL regularization. Experiments demonstrate that OWPO consistently outperforms strong baselines (e.g., MOPD, DAPO) and enables models to stably transcend suboptimal priors via iterative bootstrapping—highlighting directional awareness as a fundamental principle for scalable self-evolution.

## Limitations

OWPO has several limitations. First, our evaluation mainly focuses on math and general reasoning tasks with reliable verifiers; extending it to coding, tool-use, and interactive RLVR remains future work. Second, our theory characterizes local first-order update dynamics rather than the full global behavior of the clipped objective. Third, OWPO still depends on the reference policy and related hyperparameters, such as refresh frequency and weighting bounds, whose effects warrant further study.

## Acknowledgment

This work was supported in part by the Guangdong Grants (Grant No. 2023ZT10X075), the Natural Science Foundation of China (No. 62332002, 62425101), and the Shenzhen Science and Technology Program (KQTD20240729102051063). It was also supported by the China Postdoctoral Science Foundation under Grant Numbers BX20240013 and 2024M760113.

## Impact Statement

This paper proposes One-Way Policy Optimization (OWPO) for reinforcement learning with verifiable rewards, aiming to improve training stability and enable iterative self-improvement on verifiable reasoning tasks (e.g., math and code). This can benefit reliability and efficiency in building reasoning-oriented language models.

Broader impacts are mixed: stronger reasoning models may improve education and scientific/software tooling, but could also amplify misuse or over-reliance if outputs are deployed without adequate checking. OWPO does not involve new data collection; we recommend pairing OWPO-trained models with strong verifiers, rigorous evaluation, and responsible deployment practices in high-stakes settings.

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

# A. Proofs

This appendix provides proofs for the results in Section 4. Throughout, expectations are taken over trajectories $\tau \sim \pi_{\theta_{\text{old}}}$, i.e., the behavior policy is fixed as in standard surrogate analyses. We adopt the convention $\text{sgn}(0) = 0$.

**Definitions (for convenience).** Recall the directional deviation

$$\delta_t(\theta) \triangleq \text{sgn}(A_t) \cdot \log \frac{\pi_\theta(y_t \mid s_t)}{\pi_{\text{ref}}(y_t \mid s_t)}, \tag{11}$$

and the one-way weight (implemented with stop-gradient)

$$w_t \triangleq \text{sg}[\text{clip}(\exp(-\delta_t(\theta)), \epsilon_{\text{low}}, \epsilon_{\text{high}})], \qquad 0 < \epsilon_{\text{low}} < 1 < \epsilon_{\text{high}}. \tag{12}$$

In practice, $\text{sg}[\cdot]$ detaches the weight from backpropagation, i.e., $\nabla_\theta w_t \equiv 0$.

We also define the ratio-free policy-gradient surrogate objective

$$J_{\text{PG}}(\theta) \triangleq \mathbb{E}_{\tau \sim \pi_{\theta_{\text{old}}}} \left[ \sum_t A_t \log \pi_\theta(y_t \mid s_t) \right], \quad \nabla_\theta J_{\text{PG}}(\theta) = \mathbb{E}\left[ \sum_t A_t \nabla_\theta \log \pi_\theta(y_t \mid s_t) \right]. \tag{13}$$

## A.1. Proof of Lemma 4.1

**Lemma A.1** (Local Directional Regularization). *This proof corresponds to the local first-order update direction of the clipped PPO/DAPO surrogate. Let $\pi_{\text{ref}}$ be fixed. Under the stop-gradient implementation of $w_t$ (i.e., treating $w_t$ as constant during backpropagation), the OWPO update gradient*

$$\nabla_\theta J_{\text{OWPO}}(\theta) = \mathbb{E}_{\tau \sim \pi_{\theta_{\text{old}}}} \left[ \sum_t w_t A_t \nabla_\theta \log \pi_\theta(y_t \mid s_t) \right] \tag{14}$$

*is equal to the gradient of a regularized objective $\nabla_\theta(J_{\text{PG}}(\theta) + \Omega(\theta))$, where*

$$\Omega(\theta) = \mathbb{E}_{\tau \sim \pi_{\theta_{\text{old}}}} \left[ \sum_t |A_t| \, \psi(\delta_t(\theta)) \right], \qquad \psi(\delta) = \int_0^\delta (w(u) - 1) \, du, \tag{15}$$

*and $w(\cdot)$ denotes the scalar weight function induced by $\text{clip}(\exp(-\cdot), \epsilon_{\text{low}}, \epsilon_{\text{high}})$.*

*Proof.* We compute $\nabla_\theta \Omega(\theta)$. Since the expectation is taken over the fixed behavior policy $\pi_{\theta_{\text{old}}}$, we may move $\nabla_\theta$ inside the expectation under standard regularity assumptions (e.g., dominated convergence):

$$\nabla_\theta \Omega(\theta) = \mathbb{E}\left[ \nabla_\theta \sum_t |A_t| \, \psi(\delta_t(\theta)) \right] = \mathbb{E}\left[ \sum_t |A_t| \, \psi'(\delta_t(\theta)) \, \nabla_\theta \delta_t(\theta) \right]. \tag{16}$$

Next, because $\pi_{\text{ref}}$ is fixed,

$$\nabla_\theta \delta_t(\theta) = \nabla_\theta(\text{sgn}(A_t) \left[ \log \pi_\theta(y_t \mid s_t) - \log \pi_{\text{ref}}(y_t \mid s_t) \right]) \tag{17}$$

$$= \text{sgn}(A_t) \, \nabla_\theta \log \pi_\theta(y_t \mid s_t). \tag{18}$$

By definition, $\psi'(\delta) = w(\delta) - 1$. Substituting this relationship and the expression for $\nabla_\theta \delta_t(\theta)$ back into the equation for $\nabla_\theta \Omega(\theta)$, and letting $w_t = w(\delta_t(\theta))$, we obtain:

$$\nabla_\theta \Omega(\theta) = \mathbb{E}\left[ \sum_t |A_t| \, (w(\delta_t(\theta)) - 1) \, \text{sgn}(A_t) \, \nabla_\theta \log \pi_\theta(y_t \mid s_t) \right]. \tag{19}$$

Using the identity $|x| \, \text{sgn}(x) = x$ (with $\text{sgn}(0) = 0$) and letting $w_t = w(\delta_t(\theta))$, this simplifies to:

$$\nabla_\theta \Omega(\theta) = \mathbb{E}\left[ \sum_t (w_t - 1) \, A_t \, \nabla_\theta \log \pi_\theta(y_t \mid s_t) \right]. \tag{20}$$

Finally, adding $\nabla_\theta J_{\mathrm{PG}}(\theta) = \mathbb{E}[\sum_t A_t \nabla_\theta \log \pi_\theta]$ yields

$$\nabla_\theta \big( J_{\mathrm{PG}}(\theta) + \Omega(\theta) \big) = \mathbb{E}\left[ \sum_t (1 + w_t - 1)\, A_t\, \nabla_\theta \log \pi_\theta(y_t \mid s_t) \right] \tag{21}$$

$$= \mathbb{E}\left[ \sum_t w_t\, A_t\, \nabla_\theta \log \pi_\theta(y_t \mid s_t) \right] = \nabla_\theta J_{\mathrm{OWPO}}(\theta), \tag{22}$$

which completes the proof.

**Remark on Smoothness.** The weight function $w(\delta) = \mathrm{clip}(\exp(-\delta), \epsilon_{\mathrm{low}}, \epsilon_{\mathrm{high}})$ is continuous globally (despite having kinks at the clipping thresholds). Consequently, by the Fundamental Theorem of Calculus, the potential function $\psi(\delta)$ is continuously differentiable ($\mathcal{C}^1$) everywhere, with $\psi'(\delta) = w(\delta) - 1$ well-defined for all $\delta \in \mathbb{R}$. $\qquad\square$

### A.2. Proof of Proposition 4.2

**Proposition A.2** (Properties of One-way Dynamics). *Assuming* $0 < \epsilon_{\mathrm{low}} < 1 < \epsilon_{\mathrm{high}}$, *the effective progress coefficient*

$$g_{\mathrm{OWPO}}(\delta) \triangleq w(\delta) = \mathrm{clip}(\exp(-\delta), \epsilon_{\mathrm{low}}, \epsilon_{\mathrm{high}}) \tag{23}$$

*satisfies: (i) **Non-reversal:** $\forall \delta \in \mathbb{R}$, $g_{\mathrm{OWPO}}(\delta) \geq \epsilon_{\mathrm{low}} > 0$; (ii) **Asymmetry:** $g_{\mathrm{OWPO}}(\delta) > 1$ for $\delta < 0$, and $g_{\mathrm{OWPO}}(\delta) \in [\epsilon_{\mathrm{low}}, 1)$ for $\delta > 0$.*

*Proof.* By definition,

$$g_{\mathrm{OWPO}}(\delta) = \max \big( \epsilon_{\mathrm{low}}, \min \big( \epsilon_{\mathrm{high}}, e^{-\delta} \big) \big). \tag{24}$$

**(i) Non-reversal.** Since $e^{-\delta} > 0$ for all $\delta \in \mathbb{R}$ and $\epsilon_{\mathrm{low}} > 0$, clipping implies

$$g_{\mathrm{OWPO}}(\delta) \geq \epsilon_{\mathrm{low}} > 0, \quad \forall \delta \in \mathbb{R}. \tag{25}$$

Hence the coefficient never changes sign and cannot induce force reversal.

**(ii) Asymmetry.** If $\delta < 0$, then $-\delta > 0$ and thus $e^{-\delta} > 1$. Because $\epsilon_{\mathrm{high}} > 1$, the clipped value satisfies $g_{\mathrm{OWPO}}(\delta) \in (1, \epsilon_{\mathrm{high}}]$, hence $g_{\mathrm{OWPO}}(\delta) > 1$. If $\delta > 0$, then $e^{-\delta} \in (0, 1)$ and since $0 < \epsilon_{\mathrm{low}} < 1$, the clipped value satisfies $g_{\mathrm{OWPO}}(\delta) \in [\epsilon_{\mathrm{low}}, 1)$. (The boundary case $\delta = 0$ gives $g_{\mathrm{OWPO}}(0) = 1$ under $0 < \epsilon_{\mathrm{low}} < 1 < \epsilon_{\mathrm{high}}$.) $\qquad\square$

### A.3. Proof of Corollary 4.3

**Corollary A.3** (Variance Bounding). *Let* $G_t \triangleq A_t \nabla_\theta \log \pi_\theta(y_t \mid s_t)$ *denote the raw gradient estimator. Then the second moment of the weighted gradient satisfies*

$$\epsilon_{\mathrm{low}}^2\, \mathbb{E}\big[\|G_t\|^2\big] \leq \mathbb{E}\big[\|w_t\, G_t\|^2\big] \leq \epsilon_{\mathrm{high}}^2\, \mathbb{E}\big[\|G_t\|^2\big]. \tag{26}$$

*Moreover, conditioned on beneficial deviations ($\delta_t > 0$), we have* $\mathbb{E}[\|w_t G_t\|^2 \mid \delta_t > 0] \leq \mathbb{E}[\|G_t\|^2 \mid \delta_t > 0]$.

*Proof.* By construction, $w_t \in [\epsilon_{\mathrm{low}}, \epsilon_{\mathrm{high}}]$ almost surely, hence

$$\epsilon_{\mathrm{low}}^2 \leq w_t^2 \leq \epsilon_{\mathrm{high}}^2. \tag{27}$$

Multiplying by the nonnegative random variable $\|G_t\|^2$ yields

$$\epsilon_{\mathrm{low}}^2\, \|G_t\|^2 \leq \|w_t G_t\|^2 \leq \epsilon_{\mathrm{high}}^2\, \|G_t\|^2 \quad \text{almost surely.} \tag{28}$$

Taking expectations on both sides gives

$$\epsilon_{\mathrm{low}}^2\, \mathbb{E}\big[\|G_t\|^2\big] \leq \mathbb{E}\big[\|w_t\, G_t\|^2\big] \leq \epsilon_{\mathrm{high}}^2\, \mathbb{E}\big[\|G_t\|^2\big]. \tag{29}$$

For the conditional statement, note that by Proposition 4.2, $\delta_t > 0 \Rightarrow w_t \in [\epsilon_{\text{low}}, 1)$, hence $w_t^2 \leq 1$ on the event $\{\delta_t > 0\}$. Therefore,

$$\|w_t G_t\|^2 \leq \|G_t\|^2 \quad \text{almost surely conditioned on } \delta_t > 0, \tag{30}$$

and taking conditional expectations yields

$$\mathbb{E}\big[\|w_t G_t\|^2 \mid \delta_t > 0\big] \leq \mathbb{E}\big[\|G_t\|^2 \mid \delta_t > 0\big]. \tag{31}$$

$\square$

## B. Experimental Setup

In this section, we provide comprehensive details regarding our experimental infrastructure, training configurations, and evaluation protocols to ensure reproducibility.

### B.1. Framework and Dataset

We implement all algorithms using the **VERL** (Sheng et al., 2025) framework. For the training dataset, we utilize **dapo-math-17k** across all main experiments. This choice ensures a fair and controlled comparison with the baseline methods.

### B.2. Model Architectures and Constraints

We evaluate our method across two representative model scales and architectures: **Qwen2.5-Math-7B** and **Qwen3-8B-Base**. To accommodate the varying reasoning capabilities and context window requirements of these models, we enforce specific maximum response length constraints during training: we set a maximum length of **8k** tokens for Qwen2.5-Math-7B and extend this to **20k** tokens for Qwen3-8B-Base to fully leverage its long-context reasoning potential.

### B.3. Training Implementation Details

Our training configuration strictly follows the settings of the baselines (DAPO) to isolate the contribution of our method.

**General Optimization Settings.** Unless otherwise specified, we train with a global batch size of **512**. We employ gradient accumulation steps to manage memory, typically setting the mini-batch size to **32** with **16** accumulation steps. The learning rate is fixed at $10^{-6}$. Following the standard DAPO configuration, we exclude both KL divergence and entropy regularization losses from the primary objective, unless explicitly stated for specific baselines.

**Baseline Descriptions.** To rigorously evaluate the effectiveness of OWPO, we compare it against a spectrum of state-of-the-art RLVR and distillation methods:

- **GRPO (Group Relative Policy Optimization):** A value-free RL algorithm that estimates advantages via group sampling. It normalizes rewards against group statistics to reduce variance without requiring a learned value function.

- **DAPO (Dynamic Sampling Policy Optimization):** An advanced variant of GRPO that introduces asymmetric clipping ($\epsilon_{high} = 0.28, \epsilon_{low} = 0.2$) and a dynamic sampling mechanism. It stabilizes training by ensuring a balanced ratio of positive and negative samples within each group.

- **Sym-DAPO (Symmetric KL Regularization):** This baseline introduces a standard reference policy $\pi_{\text{ref}}$ to the vanilla DAPO. Unlike OWPO, it applies a **symmetric** KL divergence penalty (or equivalent symmetric clipping constraint) towards $\pi_{\text{ref}}$ regardless of whether the deviation leads to higher rewards. This serves as the primary control group to demonstrate the limitations of "direction-agnostic pull-back."

- **OPD (On-Policy Distillation):** A distillation-based approach (also known as GKD) where the student policy is supervised by a teacher (reference) policy on the student's own generated trajectories. It minimizes the reverse KL divergence $\mathbb{D}_{\text{KL}}(\pi_\theta \| \pi_{\text{ref}})$ to ensure the student stays close to the teacher's distribution.

- **MOPD (Multi-teacher On-Policy Distillation):** A hybrid framework that formulates distillation as an RL problem. It utilizes importance sampling with clipped ratios and constructs a hybrid advantage that combines teacher log-probabilities with outcome rewards, aiming to balance dense supervision with correctness verification.

## B.4. Hyperparameters

For our proposed **OWPO**, we introduce specific hyperparameters governing the asymmetric trust region. The dynamic weight $w_t$ is clipped within the range $[0.8, 1.2]$ (i.e., $\epsilon_{\text{low}} = 0.8$, $\epsilon_{\text{high}} = 1.2$) to prevent excessive variance while allowing sufficient modulation. For the iterative self-evolution process, we update the reference policy $\pi_{\text{ref}}$ every **80 steps** (approximate to one epoch of the training set) to facilitate the "Ratchet Effect."

*Table 4.* Hyperparameter sensitivity analysis on the AIME24 benchmark using Qwen2.5-Math-7B. We report Pass@1 and Pass@16 accuracy. The selected hyperparameters used in our main experiments are highlighted in **bold**.

| Method | Hyperparameter | AIME24 | |
|---|---|---|---|
| | | **Pass@1** | **Pass@16** |
| Sym-DAPO ($\beta$) | $10^{-3}$ | **35.00** | **51.73** |
| | $10^{-2}$ | 33.54 | 58.14 |
| | $10^{-1}$ | 28.44 | 56.14 |
| MOPD ($\alpha$) | 0.1 | 36.46 | 54.13 |
| | 0.5 | 36.25 | 53.76 |
| | 1.0 | 36.67 | 58.47 |
| | 5.0 | **38.85** | **60.28** |
| | 10.0 | 38.23 | 58.81 |
| | 50.0 | 31.98 | 51.72 |

**Baseline Hyperparameters.** To ensure a rigorous comparison, we carefully tuned the key hyperparameters for the baseline methods. We utilize **AIME24** as our validation set; therefore, all optimal hyperparameters were selected based on the model's performance on this benchmark (detailed sensitivity analysis is provided in Table 4):

- **GRPO:** We adopt the default configuration from DeepSeek-R1, setting the KL penalty coefficient to $\beta = 10^{-3}$.

- **Sym-DAPO:** For the symmetric KL regularization, we performed a hyperparameter search for the penalty coefficient $\beta$ towards the reference policy over the range $\{10^{-1}, 10^{-2}, 10^{-3}\}$. As shown in Table 4, we observed that $\beta = 10^{-3}$ yielded the optimal Pass@1 performance on AIME24 and selected it for our experiments.

- **MOPD:** Regarding the hybrid advantage formulation (see Eq. 4: $\hat{A}_{\text{MOPD}} = \text{sg}[\dots] + \alpha \hat{A}_{\text{ORM}}$), we conducted a grid search for the balancing coefficient $\alpha$ across the set $\{0.1, 0.5, 1, 5, 10, 50\}$. Our empirical results on the validation set indicated that $\alpha = 5$ achieves the best balance, and thus we adopted this value.

## B.5. Evaluation Protocol

We conduct a rigorous zero-shot evaluation across multiple benchmarks using a consistent sampling strategy with temperature $T = 1.0$ and top-$p = 0.7$.

- **Mathematical Reasoning Benchmarks (AIME 24, AIME 25, AMC):** For these specialized mathematical reasoning datasets, we sample $k = 32$ completions for each problem. We report both **Pass@1** and **Pass@16** scores to evaluate the model's precision and robustness.

- **General Reasoning Benchmarks (GPQA, MMLU-Pro):** To assess the generalizability of the learned reasoning patterns, we evaluate on GPQA (STEM-focused) and MMLU-Pro. For these broader tasks, we sample $k = 8$ completions per problem and report the average accuracy.

# C. Additional Results: Pass@16 Extension

Table 2 in the main paper reports Pass@1 under both the *best* checkpoint (selected by AIME24) and the *converged* checkpoint (after training stabilizes). To better characterize robustness under multi-sample inference, we provide an extended view that

*Table 5.* **Best performance comparison (Pass@1 vs. Pass@16).** We report the performance of the best checkpoint across different RL methods on Qwen-2.5 and Qwen-3 base models.

| Model | Method | AIME24 | | AIME25 | | AMC | | Average | |
|---|---|---|---|---|---|---|---|---|---|
| | | Pass@1 | Pass@16 | Pass@1 | Pass@16 | Pass@1 | Pass@16 | Pass@1 | Pass@16 |
| Qwen-2.5-Math-7B-Base | GRPO | 32.08 | 50.24 | 11.77 | 24.56 | 67.39 | 81.24 | 37.08 | 52.01 |
| | DAPO | 37.19 | 58.00 | 15.62 | 30.60 | 68.86 | **89.59** | 40.56 | 59.40 |
| | Sym-DAPO | 35.00 | 51.73 | 16.04 | 30.23 | 67.58 | 86.56 | 39.54 | 56.17 |
| | MOPD | 38.85 | **60.28** | 15.93 | **33.08** | 68.56 | 87.67 | 41.11 | **60.34** |
| | OPD | 38.12 | 55.06 | 15.21 | 29.14 | 68.59 | 87.75 | 40.64 | 57.32 |
| | OWPO | **41.67** | 55.84 | **17.92** | 29.20 | **72.14** | 87.41 | **43.91** | 57.48 |
| Qwen-3-8B-Base | GRPO | 31.67 | 61.98 | 23.54 | 50.91 | 67.73 | 86.89 | 40.98 | 66.59 |
| | DAPO | 36.67 | 71.41 | 27.39 | 48.98 | 71.88 | 89.87 | 45.31 | 70.09 |
| | Sym-DAPO | 36.04 | 69.98 | 26.88 | 48.16 | 70.74 | 88.98 | 44.55 | 69.04 |
| | MOPD | 37.80 | 74.68 | 29.38 | 52.25 | 72.03 | 89.83 | 46.40 | 72.25 |
| | OPD | 38.23 | 67.48 | 27.81 | 50.90 | 71.16 | 88.17 | 45.73 | 68.85 |
| | OWPO | **44.38** | **76.34** | **33.54** | **52.41** | **77.79** | **90.48** | **51.90** | **73.08** |

*Table 6.* **Converged performance comparison (Pass@1 vs. Pass@16).** We report the performance of the converged checkpoint (Last) across different RL methods on Qwen-2.5 and Qwen-3 base models.

| Model | Method | AIME24 | | AIME25 | | AMC | | Average | |
|---|---|---|---|---|---|---|---|---|---|
| | | Pass@1 | Pass@16 | Pass@1 | Pass@16 | Pass@1 | Pass@16 | Pass@1 | Pass@16 |
| Qwen-2.5-Math-7B-Base | GRPO | 29.79 | 48.38 | 12.60 | 26.99 | 64.87 | 86.67 | 35.75 | 54.01 |
| | DAPO | 31.88 | 48.84 | 17.29 | 32.27 | **77.86** | 89.05 | 42.34 | 56.72 |
| | Sym-DAPO | 32.81 | **58.01** | 15.63 | 32.52 | 71.58 | **89.53** | 40.01 | **60.02** |
| | MOPD | 37.19 | 57.30 | 16.98 | 34.14 | 62.91 | 84.71 | 39.03 | 58.72 |
| | OPD | 35.52 | 52.20 | 16.56 | **37.38** | 69.05 | 88.02 | 40.38 | 59.20 |
| | OWPO | **39.90** | 54.10 | **18.44** | 31.18 | 74.02 | 87.29 | **44.12** | 57.52 |
| Qwen-3-8B-Base | GRPO | 29.27 | 58.84 | 20.42 | 45.02 | 66.98 | 89.33 | 38.89 | 64.40 |
| | DAPO | 34.58 | 70.63 | 30.63 | **52.35** | 73.91 | 89.72 | 46.37 | 70.90 |
| | Sym-DAPO | 33.23 | 63.64 | 24.79 | 51.33 | 69.95 | 88.71 | 42.66 | 67.89 |
| | MOPD | 36.04 | 71.17 | 27.19 | 49.84 | 72.82 | 89.86 | 45.35 | 70.29 |
| | OPD | 35.42 | 67.36 | 26.56 | 49.52 | 70.22 | 88.00 | 44.07 | 68.29 |
| | OWPO | **42.19** | **73.20** | **31.67** | 49.24 | **78.36** | **91.40** | **50.74** | **71.28** |

additionally includes Pass@16. Specifically, Table 5 reports the **best** checkpoint performance (Pass@1 vs. Pass@16), and Table 6 reports the corresponding **converged** checkpoint performance.

**Key observation.** Across both base models and across benchmarks (AIME24/AIME25/AMC), OWPO improves Pass@1 while maintaining competitive Pass@16. This indicates that the gains of OWPO are not merely a narrow improvement at the top-1 sample; rather, the method preserves multi-sample success rates that reflect broader coverage of correct reasoning trajectories. In particular, OWPO's one-way reweighting (accelerated correction for harmful deviations and gain locking for beneficial ones) stabilizes training and reduces variance, which is consistent with retaining strong Pass@16 performance even as Pass@1 improves.

**Best vs. Converged checkpoints.** Comparing Table 5 and Table 6, OWPO exhibits strong *converged* Pass@1 while remaining competitive in Pass@16, suggesting that the method mitigates late-stage regression without sacrificing sampling robustness. Overall, these results complement Table 2 by showing that OWPO simultaneously improves *precision* (Pass@1) and preserves *robustness* under additional sampling (Pass@16).

