# OpenReview forum: "One-Way Policy Optimization for Self-Evolving LLMs"
_ICML.cc/2026/Conference — ICML 2026 regular_

### Official Review · Reviewer_WDtx · 2026-03-12

**Soundness:** 3
**Presentation:** 3
**Significance:** 3
**Originality:** 2
**Overall Recommendation:** 4
**Confidence:** 3

**Summary:**

This paper studies reference-guided RLVR for LLM reasoning and argues that standard reference-based constraints can conflict with the verifier’s reward-improving direction. The core proposal, OWPO, reweights PPO-style updates using a token-level factor derived from the sign of the sequence-level advantage and the log-ratio between the current and reference policies. The intended effect is asymmetric: amplify updates for “inferior” deviations and downweight updates for “superior” deviations, while periodically refreshing the reference policy to enable iterative self-evolution. Empirically, the paper reports gains over several benchmarks compared with strong baselines.

**Compliance With Llm Reviewing Policy:**

Affirmed.

**Final Justification:**

The rebuttal addressed my concerns so I maintain my positive score.

**Key Questions For Authors:**

See Weakness.

**Limitations:**

Yes

**Strengths And Weaknesses:**

**Strength**

The paper addresses a real and important issue in RLVR: how to benefit from a reference policy without letting the reference override reward-improving directions. The high-level idea of decoupling direction from magnitude is intuitive, and the method is simple enough that others could plausibly build on it. The empirical comparison is reasonably, and the iterative self-evolution experiment is a useful stress test. The paper also includes ablations on asymmetric weighting, gain locking, and accelerated alignment, which is helpful.

**Weakness**
(1) The Pass@1 improvements are encouraging, but the Pass@16 story is more mixed than the main text suggests: OWPO is often competitive, but not uniformly best or the gain is limited, and in some converged Qwen-2.5 settings its Pass@16 average is below several baselines. Could the authors explain the potential reasons?

(2) The tested model famility and benchmarks are limited. Qwen is the only tested model class. Since AIME24 is selected as the validation set,  AIME25 AMC are the only two real test sets but both of them have small size. What is the performance on MATH500 and Olympicbench?

---

> ### Author Rebuttal · Authors · 2026-03-30
>
> We sincerely appreciate your constructive feedback. Below we address each concern in detail.
>
> > Q1: Pass@1 vs. Pass@16
>
> We appreciate this sharp observation. We agree that the Pass@16 story is more nuanced than the Pass@1 story, especially for the Qwen-2.5 setting. OWPO is consistently strongest on Pass@1, while its Pass@16 is better described as competitive rather than uniformly best, particularly at convergence for Qwen-2.5.
>
> Our interpretation is that this reflects a **precision–coverage tradeoff** induced by the optimization target. OWPO is explicitly designed to preserve and consolidate reward-improving trajectories: for Superior deviations, the gain-locking mechanism shrinks the update magnitude to reduce variance and protect already beneficial behaviors; for Inferior deviations, accelerated alignment pushes the policy more strongly toward correction. This combination is highly effective for improving top-probability precision, which is consistent with the **strong Pass@1** gains we observe. At the same time, OWPO does not include any explicit diversity or coverage objective across multiple sampled trajectories, so it is not optimized to maximize Pass@16 uniformly in the same way.
>
> This interpretation is also consistent with **recent literature**. In particular, recent work [1] has argued that RLVR often improves sampling efficiency at small k without necessarily expanding broader reasoning coverage beyond the base distribution, which helps explain why gains can be stronger on Pass@1 than on larger-pass@k metrics. If you want an even more direct citation on the mechanism, you can additionally note that recent analysis of GRPO-style updates attributes weaker large-k performance to **distribution sharpening**, where high-probability successful trajectories are reinforced while rarer correct trajectories are under-preserved.
>
> We will revise the text to make this nuance explicit and avoid overstating the Pass@16 claim.
>
> ---
>
> > Q2: Evaluation on larger benchmarks
>
> To address the your concern more directly, we have conducted **additional evaluations on MATH500 and OlympiadBench** and provide the new results here: **https://postimg.cc/ZBjLZ1xr**. These additional results show that OWPO remains **strong on larger and more challenging mathematical benchmarks**, which strengthens the evidence that the gains are not merely an artifact of the relatively small AIME/AMC evaluation sets.
>
> We also agree that the current model-family coverage remains limited, since our experiments are conducted on Qwen-2.5 and Qwen-3 base models. Due to the rebuttal time constraint, we prioritized expanding the benchmark coverage first.
>
> ---
>
> Reference:
>
> [1] Does Reinforcement Learning Really Incentivize Reasoning Capacity in LLMs Beyond the Base Model?

---

> > ### Author Rebuttal · Reviewer_WDtx · 2026-04-05
> >
> > Thanks for your response. My concern is resolved and I will maintain my positive score.

---

### Official Review · Reviewer_SXCE · 2026-03-13

**Soundness:** 4
**Presentation:** 4
**Significance:** 4
**Originality:** 3
**Overall Recommendation:** 5
**Confidence:** 2

**Summary:**

Reinforcement Learning with Verifiable Rewards (RLVR) has emerged as a key technique for enhancing the reasoning capabilities of Large Language Models (LLMs) in recent years. In RLVR, a binary verifier is introduced to provide additional training signals. Meanwhile, existing approaches primarily employ a reference policy to stabilize training. However, the verifier and the reference policy may have conflicting optimization directions. To address this issue, this paper proposes One-Way Policy Optimization (OWPO), where the verifier determines the update direction while the reference policy adjusts the update magnitude. Furthermore, this work extends OWPO into an iterative framework to achieve continuous self-evolution. Experimental results demonstrate the superiority of OWPO.

**Compliance With Llm Reviewing Policy:**

Affirmed.

**Final Justification:**

My concerns have been addressed. Therefore, I maintain my original score.

**Key Questions For Authors:**

1. What performance gain can be achieved when employing only the "Self-Evolution via Ratchet Effect" mechanism?
2. Does the theoretical analysis remain valid when the condition "at the start of an iteration" is not satisfied? I can understand the necessity of setting such a condition, but I am also willing to see the authors' understanding of this condition.

**Limitations:**

yes

**Strengths And Weaknesses:**

Strengths:

Soundness: The effectiveness of OPPO has been demonstrated from both theoretical and empirical perspectives. Comprehensive experiments are conducted in this work, including thorough comparisons with representative RLVR and distillation methods, as well as ablation studies that validate the effectiveness of each component. Furthermore, a hyperparameter analysis is presented, which facilitates the generalization and improvement of the proposed approach.

Presentation: This paper is well-organized and clearly presented, with logical coherence throughout. The motivation is described in a clear and accessible manner, and the mathematical notations and formulations adhere to established conventions.

Significance: RLVR has emerged as a key technique for enhancing the reasoning capabilities of LLMs. However, the verifier and the reference policy may exhibit conflicting optimization directions. This paper proposes a one-way update approach, which holds promise as an effective technique for addressing this challenge.

Originality: Based on the verifier’s signal and the deviation from the reference policy, this paper categorizes the policy behaviors into four cases, which are further grouped into two classes:
1. Inferior Deviations : the policy lags behind the reference (e.g., lower confidence on correct tokens or higher confidence on wrong ones);
2. Superior Deviations: the policy outperforms the reference (e.g., higher confidence on correct tokens).
Different loss weights are assigned to these two classes through the defined directional deviation (Eq. 5), as shown in Eq. 7.
Furthermore, the reference $π_{ref}$ is periodically updated to the policy $π_{θ}$, which supports a unidirectional climb toward higher rewards.
The proposed method is straightforward and effective.

Weaknesses:

Please refer to “Questions”.

---

> ### Author Rebuttal · Authors · 2026-03-30
>
> We sincerely thank the reviewer for the very positive assessment of our paper and for recognizing the soundness, clarity, and potential impact of our work. We are highly encouraged by your support. Below, we address your two questions directly.
>
> > Q1: What performance gain can be achieved by the “Self-Evolution via Ratchet Effect” alone?
>
> Thank you for this important question. **Sec. 5.3 provides a clean estimate of this effect** because OWPO and MOPD use the same hard-swap iterative schedule: at the end of each stage, the best cpt is frozen and used as the reference for the next stage. This ensures that both methods benefit from the identical “ratchet” schedule, so the MOPD trajectory gives a direct estimate of the gain achievable from iterative reference refreshing without OWPO’s one-way reweighting. Under this protocol, starting from a suboptimal prior of about 30% on AIME24, MOPD improves to about 37% after 6 iterations, indicating that the **ratchet mechanism alone** already provides a solid gain of roughly +7 points.
>
> At the same time, Sec. 5.3 also shows that the ratchet mechanism alone is not sufficient for the best outcome. Under the **same iterative protocol**, OWPO reaches about 40% within only 4 iterations, whereas MOPD saturates around 37%. We therefore view the ratchet effect as the self-evolution scaffold, while OWPO’s asymmetric one-way weighting makes each stage of that climb more effective by preserving superior deviations and avoiding the regularization conflict that limits symmetric methods.
>
> ---
>
> > Q2: Does the theoretical analysis remain valid beyond the “start of an iteration” condition?
>
> Thank you for this insightful question. Our view is that the answer is **yes in spirit, but with an important distinction between the exact local interpretation and the global practical behavior**.
>
> The analysis in Sec. 4 intentionally focuses on the local optimization landscape at the start of an iteration, where $\pi_\theta \approx \pi_{\text{old}}$ and thus $r_t(\theta)\approx 1$. Under this condition, Lemma 4.1 provides a clean interpretation of OWPO as an asymmetric directional regularizer. We use this regime because it allows us to expose the essential mechanism of OWPO in the simplest and most rigorous way.
>
> When this condition is no longer satisfied exactly later in training, we would view the equivalence in Lemma 4.1 as a local approximation rather than a literal global identity. However, the core one-way property of OWPO still remains. The one-way weight
> $w_t$ is always positive and bounded, so it acts only as a modulation of update magnitude and does not by itself introduce a sign flip relative to the underlying PPO/DAPO surrogate update. In other words, once we move away from the exact start-of-iteration regime, what becomes approximate is the clean regularization interpretation, not the intended one-way behavior of “direction from verifier, magnitude from reference.”
>
> Empirically, this local assumption also remains highly relevant in practice. Because PPO/DAPO clipping constrains the policy from drifting too far from $\pi_{\text{old}}$, the token importance ratio remains concentrated near 1 during training. We tracked the distribution of $r_t(\theta)$ throughout training, and the resulting histograms show that the ratio stays sharply centered around 1, with low variance across the vast majority of updates **(See fig link: https://postimg.cc/K1LwF8kH)**. This empirical evidence supports that the local condition used in Sec. 4 is not merely a proof convenience, but a good approximation to the practical optimization regime as well.
>
> ---
>
> We thank the reviewer again for the encouraging assessment and for these helpful questions.

---

> > ### Author Rebuttal · Reviewer_SXCE · 2026-04-03
> >
> > I thank the authors for their clarification. My concerns have been addressed. The answer to Question 1 explains the effectiveness of the Ratchet Effect. The answer to Question 2 explains the reasonableness of the condition for the theoretical analysis.

---

> > > ### Author Response · Authors · 2026-04-03
> > >
> > > We appreciate your positive score and thoughtful review. We are glad that our rebuttal addressed your questions.

---

### Official Review · Reviewer_Uj5N · 2026-03-13

**Soundness:** 2
**Presentation:** 3
**Significance:** 2
**Originality:** 2
**Overall Recommendation:** 3
**Confidence:** 4

**Summary:**

The paper proposes a weighting scheme for the updates in GRPO. This weight strength is modulated by the alignment between the sign of the advantage and log likelhood ratio of the policy to the reference. Authors consider different regimes of these sign alignment : Accelerated Alignment for Inferior deviations (where the policy lags behind the reference) and Gain Locking for Superior deviations (where the policy surpasses the reference). The reference model is also moved along the training. Experiments are shown on reasoning tasks.

**Compliance With Llm Reviewing Policy:**

Affirmed.

**Final Justification:**

I appreciate the additional experiments the reviewer provided, I still think the rational behind the method need further investigation to be ready for publication.

**Key Questions For Authors:**

My main question:
- How does the method improve the sparsity of the reward in GRPO ?
- The arguments presented to support the method and the theoretical presentation are not convincing, seeing two importance sampling weights in the update may indicate that another reasoning from first principle is possible and I may be over-interpreting please correct me if I am:

maybe one way of thinking of the proposed method : 1) if $ m=A \log (\pi/\pi_{ref}) > 0$ , consider samples  from a geometric mean of $\pi_{ref}^{\gamma}\pi_{old}^{1-\gamma}$, and then it is natural to have the importance sampling  $\pi / (\pi_{ref}^{\gamma}\pi_{old}^{1-\gamma})$, and one can reason then about splitting it to $sg (\pi^{\gamma}/ (\pi_{ref}^{\gamma})$ and putting inside clipping $(\pi/pi_{old})^{1-\gamma}$. This corresponds to using a (reverse KL) regularizer for m>0.  $\gamma  E_{\pi }1_{m>0}\log (\pi || \pi_{ref})+ (1-\gamma) E_{\pi} 1_{m>0}\log(\pi || \pi_{\theta_{old}})$  2) if $A log (\pi/pi_{ref}) < 0$, one can try to use a forward KL (integrated only on $m <0$), and then here one considers samples from the arithmetic mean of $pi_{ref}$ and $\pi_{old}$,  and then the natural ratio likelhood in that case is $\pi/ (\gamma \pi_{ref} + (1-\gamma) \pi_{old})$ . see for e.g KL-Regularized RLHF with Multiple Reference Models: Exact Solutions and Sample Complexity

- Can you please add variances in all experiments so we get statistical significance

**Limitations:**

- theory is a bit handwavy
- Experiments lack statistical significance

**Strengths And Weaknesses:**

## Soundness
* The weighting by sg (\pi/pi_{ref}) or sg(\pi_{ref}/\pi) that is modulated by  \delta is reasonable  but it sounds more of a hack than a principled  method. The idea of switching between forward and backward KL depending on the the advantage sign is reasonable , but implementing it in the advantage cost as a likelihood ratio  weighting evaluated on sample from old policy and linking that to  comparing reference to current policy is a bit handwavy.
* Figure 2 has typos  in gain locking pi_{ref}/pi_{ref}, same for encourage alignment
* The paper claims that this method helps with the sparsity of the reward, but it is not clear to me that this methods creates success in the verifier, sparsity in context of RLVR refers to all successes or all failures which does not move the training much.

## Presentation

Overall the paper is well written and conveys the ideas with illustrations .

## Significance

While the proposed method is a bit handwavy , the paper shows some promising results nevertheless they don't show any statistical significance to know if the gains are real

## Originality
There has been a lot of works on stabilizing GRPO and this papers contributes in this direction.

---

> ### Author Rebuttal · Authors · 2026-03-30
>
> We sincerely appreciate your constructive feedback. Below we address each concern in detail.
>
> ---
>
> > Q1 & Soundness W3: Clarification on "Reward Sparsity"
>
> We thank the reviewer for raising this point.  We clarify that the **“reward sparsity” relevant to our paper is not the group-level advantage degeneracy** addressed by DAPO, but the **token-level optimization** coarseness caused by sparse sequence-level verifier rewards[1, 2, 3, 4].
>
> First, there is **group-level sparsity** in GRPO-style training: when a sampled group contains all-success or all-failure rollouts, the normalized advantage can become degenerate or uninformative. In our framework, this issue is already handled by the **DAPO foundation inherited in Sec. 2**, where dynamic sampling explicitly enforces mixed success/failure within a batch. OWPO is built on top of this stabilized setting.
>
> Second, there is **token-level credit sparsity** [1, 2, 3, 4]. In RLVR for reasoning, the verifier typically provides only a **sequence-level binary outcome**. GRPO-style training then converts this into a **single group-normalized advantage** for the whole sampled response, and this same scalar is applied uniformly to every token in the trajectory. As a result, all reasoning steps in a long chain-of-thought receive the same update direction and base magnitude, even though different tokens may contribute very differently to the final outcome. This is the token-level sparsity/credit-assignment issue we aim to address.
>
> OWPO addresses this by introducing **token-dependent weighting** from the reference policy. Concretely, while the verifier-provided advantage still determines the optimization direction, OWPO computes a **token-wise** directional deviation relative to the reference policy and uses it to reweight each token’s update asymmetrically. In this way, OWPO converts a uniform sequence-level learning signal into a more informative token-level optimization signal.
>
> ---
>
> > Q2 & Soundness W1: Theoretical grounding and relation to KL-based views
>
> We thank the reviewer for **this insightful perspective**. Our main point is that this is a useful KL-style interpretation of OWPO, but not an algebraically equivalent derivation of our objective. OWPO remains a PPO update with the standard importance ratio with respect to $\pi_{\mathrm{old}}$, and introduces only a sign-conditioned stop-gradient token weight defined relative to $\pi_{\mathrm{ref}}$.
>
> Concretely, for positive advantages the weight is proportional to $\pi_{\mathrm{ref}}/\pi_\theta$, while for negative advantages it is proportional to $\pi_\theta/\pi_{\mathrm{ref}}$, yielding the asymmetric accelerated-alignment / gain-locking behavior within a PPO-style update.
>
> Accordingly, our theoretical claim is intentionally modest: OWPO is best understood as a practical local surrogate, rather than the exact solution of a mixed-reference KL objective. Empirically, the token importance ratios remain tightly concentrated around 1 throughout training, which is consistent with the local-regime assumption used in Sec. 4 **(see ratio histogram: https://postimg.cc/K1LwF8kH)**.
>
> We find the reviewer’s formulation **highly insightful**, and we believe it provides a valuable lens for further understanding OWPO. We will explore this mixed-reference/KL-style perspective more carefully in future work.
>
> ---
>
> > Q3 & Significance: Variance across runs
>
> To address this directly, we conducted **repeated-run** experiments with multiple random seeds on our primary benchmarks and now report the corresponding statistics. As shown in the table **(See table link: https://postimg.cc/ykT3GFnp)**, OWPO not only improves the mean performance across runs, but also consistently reduces run-to-run variance relative to DAPO.
>
> ---
>
> > Soundness W2: Typos in Figure 2
>
> Thank you for the careful reading. We have corrected the corresponding ratio expressions. These are figure-level annotation errors only and do not affect the formal objective, equations, or implementation in Sec. 3.1. The corrected figure is provided here **(See fig link: https://postimg.cc/TKV9PLdX)**.
>
> ---
> Reference:
>
> [1] Let's Verify Step by Step
>
> [2] Rewarding Progress: Scaling Automated Process Verifiers for LLM Reasoning
>
> [3] FIPO: Eliciting Deep Reasoning with Future-KL Influenced Policy Optimization
>
> [4] GTPO and GRPO-S: Token and Sequence-Level Reward Shaping with Policy Entropy

---

> > ### Author Rebuttal · Reviewer_Uj5N · 2026-04-01
> >
> > Thanks for the clarification and for the additional experiments on variance and for fixing figure 2.
> >
> > I still don't understand the method beyond the intuitions but  from the perspective of a forward/backward KL point of view and it would be interesting to understand this more and maybe to analyze empirically these questions. I encourage the authors to look more into this.

---

> > > ### Author Response · Authors · 2026-04-02
> > >
> > > We sincerely thank the reviewer for this profound suggestion. Viewing OWPO through your Forward/Reverse-KL lens **provides a useful interpretation of** our asymmetric update geometry, and we have added **additional empirical analysis** to provides a useful local interpretation.
> > >
> > > ---
> > >
> > > ### Theoretical Alignment with Your Formulas.
> > >
> > > We agree that the key quantity is $m = A \log \frac{\pi}{\pi_{\mathrm{ref}}}$. In the local regime assumed in Sec. 4, where the PPO ratio remains close to 1, $\frac{\pi}{\pi_{\mathrm{old}}}\approx 1$, your two mixed-reference branches reduce to functions of the same quantity $\log(\pi/\pi_{\mathrm{ref}})$ that drives the OWPO asymmetry.
> > >
> > > - For m>0:  $ \frac{\pi}{\pi_{\mathrm{ref}}^{\gamma}\pi_{\mathrm{old}}^{1-\gamma}} = \left(\frac{\pi}{\pi_{\mathrm{ref}}}\right)^{\gamma} \left(\frac{\pi}{\pi_{\mathrm{old}}}\right)^{1-\gamma} \approx \left(\frac{\pi}{\pi_{\mathrm{ref}}}\right)^{\gamma},$
> > > - For m<0: $ \frac{\pi}{\gamma\pi_{\mathrm{ref}}+(1-\gamma)\pi_{\mathrm{old}}} \approx \frac{q}{\gamma+(1-\gamma)q}, \qquad q:=\frac{\pi}{\pi_{\mathrm{ref}}}.$
> > >
> > > When $q\approx 1$, both branches are first-order functions of $\log q$; the correspondence is especially direct at $\gamma=1$, where both mixed references collapse to $\pi_{\mathrm{ref}}$. In this sense, we agree that your formulation captures the key local geometry of the method: a reverse-KL-like branch for $m>0$ and a forward-KL-like branch for $m<0$.
> > >
> > > ---
> > >
> > > ### Empirical Validation of the Dual-KL Dynamics
> > >
> > > Motivated by your suggestion, for $m_t := A_t \log\frac{\pi_\theta}{\pi_{\rm ref}} \equiv \delta_t$, we directly measured **token-level reverse KL and forward KL** relative to $\pi_{\mathrm{ref}}$ during OWPO training. The resulting asymmetry matches this dual-KL picture very clearly, see the figure below for a direct visualization **(fig link https://postimg.cc/4YSxQjyf)**:
> > >
> > > | Subset | Reverse KL $\mathrm{KL}(\pi_\theta\|\pi_{\mathrm{ref}})$ | Forward KL $\mathrm{KL}(\pi_{\mathrm{ref}}\|\pi_\theta) $ | More regularization like |
> > > |---|---:|---:|---|
> > > | $m_t > 0$ | **0.01974** | 0.04890 | Reverse-KL regularization |
> > > | $m_t < 0$ | 0.03807 | **0.01680** | Forward-KL regularization |
> > >
> > > Empirically, on the $m_t>0$ subset, reverse KL is smaller than forward KL, whereas on the $m_t<0$ subset, forward KL is smaller than reverse KL. This sign-flip pattern **is consistent with** the reviewer’s proposed reverse-/forward-KL interpretation. As a supplementary check, the token importance ratios also remain tightly concentrated around 1 throughout training (see ratio histogram: https://postimg.cc/K1LwF8kH), which is consistent with the local-regime assumption above.
> > >
> > > ---
> > >
> > > We sincerely thank you for all the constructive suggestions that have strengthened our paper. We remain fully available to address any remaining concerns. We hope this clarification and the new experiment fully address the concern.

---

### Official Review · Reviewer_4dtf · 2026-03-15

**Soundness:** 3
**Presentation:** 2
**Significance:** 3
**Originality:** 2
**Overall Recommendation:** 3
**Confidence:** 2

**Summary:**

This paper introduces One-Way Policy Optimization (OWPO), a reinforcement learning method designed for RL with Verifiable Rewards (RLVR) in large language models. The authors identify a limitation of existing reference-guided RL approaches: token-level constraints (e.g., KL penalties) can oppose reward-improving directions, potentially reversing gradient updates and limiting performance improvements. To address this, the paper proposes decoupling optimization direction and update magnitude. In OWPO, the verifier determines the direction of the update via the advantage signal, while the reference policy modulates only the step size through asymmetric weighting. The method distinguishes between inferior deviations (policy worse than reference) and superior deviations (policy better than reference), applying accelerated alignment or gain-locking mechanisms, respectively. The authors further introduce an iterative "ratchet" mechanism that periodically updates the reference policy to enable continuous improvement. Experiments on math reasoning benchmarks show that OWPO consistently outperforms several RLVR baselines, including GRPO, DAPO, OPD, and MOPD.

**Compliance With Llm Reviewing Policy:**

Affirmed.

**Key Questions For Authors:**

1. The method relies on a reference policy to modulate update magnitude. How sensitive is OWPO to the quality of the initial reference model? For example, does the method still perform well when the reference policy is significantly weaker or noisier?

2. Most experiments focus on mathematical reasoning benchmarks. Have the authors evaluated OWPO on other RLVR tasks (e.g., coding or tool-use tasks with verifiable rewards)? Evidence of consistent improvements across domains would strengthen the claims.

3. Does OWPO introduce additional computational cost compared to standard PPO-style RLVR methods due to token-level weighting and reference refresh steps? Clarifying the training overhead would help assess practical usability.

**Limitations:**

yes

**Strengths And Weaknesses:**

Soundness

The paper proposes a clear modification to RLVR training by introducing asymmetric token-level weighting that separates the direction of optimization from its magnitude. The formulation of the OWPO objective and its connection to directional regularization are well described, and the theoretical analysis provides intuition for why the method avoids the "force reversal" effect that can arise from symmetric KL penalties. The empirical evaluation is reasonably thorough. Experiments are conducted on two base models and multiple benchmarks, including AIME24, AIME25, AMC, GPQA, and MMLU-Pro. Results in Table 2 show consistent improvements over strong baselines, and the training curves illustrate that OWPO maintains steady improvement rather than plateauing near the reference policy. However, the theoretical analysis remains relatively high-level. Some arguments rely on simplifying assumptions, such as approximating the importance sampling ratio near 1, and the connection between the theoretical framework and the practical PPO-style implementation could be clarified further. Additionally, improvements over baselines, while consistent, are moderate in magnitude.

Presentation

The paper is generally well structured and clearly motivates the problem of directional conflicts in reference-guided RL training. Figures illustrating the difference between standard KL regularization and OWPO’s asymmetric dynamics help convey the intuition behind the method. Some parts of the methodology section introduce multiple concepts (e.g., directional deviation, asymmetric weighting, and ratchet effect) in quick succession, which can make the overall algorithm slightly difficult to follow on first reading. A clearer step-by-step description of the training procedure would improve readability.


Significance

Improving RL with verifiable rewards is an important problem for scaling reasoning abilities in large language models. Sparse binary rewards are a well-known challenge in RLVR, and the paper addresses a meaningful limitation of existing reference-guided approaches. If the proposed mechanism consistently improves stability and enables models to surpass reference policies without external teachers, it could be useful for future research on self-improving LLM training pipelines. However, the work mainly represents an incremental improvement to existing RLVR optimization strategies rather than a fundamentally new paradigm.


Originality

The key idea of decoupling optimization direction from update magnitude through asymmetric weighting is interesting and conceptually simple. While related to prior work on trust-region RL and reference-guided RLHF methods, the specific formulation of directional deviation and the gain-locking mechanism provide a new perspective on handling reference policies. In other words, OWPO remains closely related to existing PPO-style RL methods with modified weighting or clipping strategies. The novelty primarily lies in the asymmetric reweighting scheme and the associated interpretation of directional regularization.

---

> ### Author Rebuttal · Authors · 2026-03-30
>
> We sincerely appreciate your constructive feedback. Below we address each concern in detail.
> > Q1: Sensitivity to the initial reference model
>
> We already study this setting in **Sec. 5.3**. Specifically, we initialize the reference policy with a suboptimal cpt at 30.0% AIME24 rather than the best 37.19% cpt, and compare OWPO with MOPD under the same iterative hard-swap protocol. Under this controlled setting, OWPO reaches approximately 40% Pass@1 within 4 iterations, whereas MOPD saturates near 37% after 6 iterations. This suggests that OWPO is not tightly bottlenecked by a weak prior, can surpass a suboptimal reference efficiently, and does so with higher self-evolution efficiency under the same iterative schedule.
>
> ---
>
> > Q2: Evaluation beyond math reasoning
>
> We agree that coding or tool-use RLVR would provide an even stronger cross-domain test. Due to the limited rebuttal period, we are unable to build a new RL pipeline in such a different domain within the rebuttal window. To nevertheless broaden the empirical evidence, we conducted **additional evaluations (See table link https://postimg.cc/ZBjLZ1xr)** on Minerva, MATH500, and OlympicBench. Minerva contains 272 undergraduate-level STEM problems spanning diverse subjects such as **Chemistry, Biology, and Astronomy**. Across these added benchmarks, OWPO continues to outperform the baselines, which strengthens the case that the benefit is not restricted to the original AIME/AMC setting but extends to broader reasoning scenarios. This is also consistent with Table 3 in the paper, where math-trained OWPO already transfers better to GPQA and MMLU-Pro.
>
> ---
>
> > Q3: Computational cost
>
> OWPO introduces only modest additional overhead compared with standard PPO-style RLVR. Our implementation follows the same training configuration as DAPO (Appendix B.3), so the comparison is controlled. The main extra cost comes from evaluating the reference policy on on-policy samples to obtain token-level log-probabilities which is lightweight. We measured the wall-clock cost in our setup:
> | Component | Time per Step (s) | Overhead vs DAPO |
> | :- | :- | :- |
> | Standard DAPO Step | 609.8s | - |
> | Token-level Weighting (OWPO) | +17.8s | +2.9% |
> | **Total OWPO Step** | **627.6s** | **+2.9%** |
>
> Overall, this corresponds to only a small per-step overhead (+2.9% in our setup). Figure 4 in paper shows that under a comparable training budget, OWPO does not require a longer optimization horizon to realize its gains. In the iterative weak-prior setting, OWPO is also more self-evolution-efficient than MOPD, reaching ≈40% in 4 iterations versus MOPD’s ≈37% after 6 iterations.
>
> ---
>
> > On Theoretical assumptions
>
> Regarding the theoretical analysis and the local assumption $r_t(\theta)\approx 1$. Our goal in Sec. 4 is to isolate the directional effect of OWPO in the local start-of-iteration regime where $\pi_\theta \approx \pi_{\text{old}}$, under which the importance ratio satisfies $r_t(\theta)\approx 1$.
>
> To further validate that this approximation is relevant in practice, we tracked the empirical distribution of $r_t(\theta)$ during training. Following the diagnostic style **used in recent work [1]**, we add analogous histograms **(See figure link https://postimg.cc/K1LwF8kH)**. Our results show that the token-wise ratio remains sharply concentrated around 1, with the variance of $r_t(\theta)$ consistently below 0.005 in our runs. These diagnostics provide empirical support that the local assumptions used in our analysis are well aligned with the practical PPO-style implementation for the vast majority of updates.
>
> ---
>
> > On presentation
>
> We agree that the method section can be made easier to follow. The current manuscript already includes Figure 2 and Algorithm 1; We will further align them into a more explicit step-by-step pipeline that separates (i) directional deviation computation, (ii) asymmetric weighting, (iii) policy update, and (iv) periodic reference refresh.
>
> ---
>
> > On significance and originality
>
> We agree that OWPO is not intended as a fundamentally new paradigm, but rather as a **targeted and principled modification** within the PPO-style RLVR family. Our claim is that this asymmetric objective change is practically meaningful: it removes force reversal, improves self-evolution efficiency under the same iterative schedule, and yields more stable optimization. To make this point more explicit, we add **repeated-run statistics across multiple random seeds (See table link https://postimg.cc/ykT3GFnp)** , showing that OWPO not only improves mean performance but also reduces variance compared with existing baselines.
>
> ---
> Reference:
>
> [1] Soft Adaptive Policy Optimization

---

### Decision · Program_Chairs · 2026-04-30

**Decision:**

Accept (regular)

**Comment:**

I recommend acceptance here along with most reviewers.

The main objection come from reviewer Uj5N who is worried about the rigor of the derivations and would prefer a better connection to KL. I think the author's final rebuttal makes this connection a bit more clear and so I am ok recommending acceptance if the authors include this clarification in the final draft.

Otherwise, the paper builds some understandable intuitions about interference between KL regularization and RL and the empirical results are generally sound according to all reviewers.